# Engineered transcription-associated Cas9 targeting in eukaryotic cells

Gregory W. Goldberg [1] ✉, Manjunatha Kogenaru [1], Sarah Keegan[1], Max A. B. Haase[1], Larisa Kagermazova [1], Mauricio A. Arias[2], Kenenna Onyebeke[1], Samantha Adams[1], Daniel K. Beyer[1], David Fenyö[1], Marcus B. Noyes [1] ✉ & Jef D. Boeke [1,3] ✉

DNA targeting Class 2 CRISPR-Cas effector nucleases, including the well-studied Cas9 proteins, evolved protospacer-adjacent motif (PAM) and guide RNA interactions that sequentially license their binding and cleavage activities at protospacer target sites. Both interactions are nucleic acid sequence specific but function constitutively; thus, they provide intrinsic spatial control over DNA targeting activities but naturally lack temporal control. Here we show that engineered Cas9 fusion proteins which bind to nascent RNAs near a protospacer can facilitate spatiotemporal coupling between transcription and DNA targeting at that protospacer: <u>Tra</u>nscription-associated <u>C</u>as9 <u>T</u>argeting (TraCT). Engineered TraCT is enabled in eukaryotic yeast or human cells when suboptimal PAM interactions limit basal activity and when one or more nascent RNA substrates are still tethered to the actively transcribed target DNA in *cis*. Using yeast, we further show that this phenomenon can be applied for selective editing at one of two identical targets in distinct gene loci, or, in diploid allelic loci that are differentially transcribed. Our work demonstrates that temporal control over Cas9's targeting activity at specific DNA sites may be engineered without modifying Cas9's core domains and guide RNA components or their expression levels. More broadly, it establishes co-transcriptional RNA binding as a *cis*-acting mechanism that can conditionally stimulate CRISPR-Cas DNA targeting in eukaryotic cells.

Class 2 <u>C</u>RISPR-<u>as</u>sociated (Cas) effectors such as Cas9 are vastly simplifying (epi)genome editing applications in biomedical research, owing to their impressive DNA targeting specificity, facile guide RNA programmability, and tractable single-protein architecture[1–5]. In complex with guide RNAs, Cas9 proteins constitutively bind and cleave DNA at target sites known as protospacers[6–8]. Binding and cleavage occur sequentially through a coupled mechanism[9] initiated by direct protein-DNA interactions at protospacer-adjacent motifs (PAMs)[10,11], for example, 'NGG' sequences[12] recognized by the model Cas9 protein from *Streptococcus pyogenes* SF370. Following PAM

recognition, Cas9 forms an R-loop and cleaves its target once sufficient base-pairing is achieved between its guide RNA and protospacer DNA[8,9,13–17]. Mechanistic coupling between PAM and guide RNA interactions enforces sequence-specific spatial control over Cas9's DNA targeting activities; i.e., even perfectly complementary guides cannot license binding or cleavage at protospacers that lack a functional PAM. Furthermore, suboptimal PAM (subPAM) interactions may license intermediate activities, as seen with the NAG and NGA PAMs of *S. pyogenes* Cas9 that are recognized less strongly than NGG[12,18,19].

[1]Institute for Systems Genetics and Department of Biochemistry and Molecular Pharmacology, NYU Langone Health, New York, NY, USA. [2]Courant Institute of Mathematical Sciences, New York University, New York, NY, USA. [3]Department of Biomedical Engineering, NYU Tandon School of Engineering, Brooklyn, NY, USA. ✉e-mail: Gregory.Goldberg@nyulangone.org; Marcus.Noyes@nyulangone.org; Jef.Boeke@nyulangone.org

Beyond Cas9's broad utility as a tool for constitutive DNA binding and cleavage, certain applications in cells demand dynamically modulated activity that can be controlled over time by an experimentalist or via an endogenous signal. Temporal control was previously engineered by inducing Cas9 expression[20], as well as by modifying Cas9's guide RNA or protein components to ensure that they are held inactive until post-transcriptionally or post-translationally induced by chemical[21–24], optical[25–28], or enzymatic[29] inputs. Engineered Cas9 'pro-enzymes' were further shown to respond to intracellularly expressed proteases via proteolytic activation[29], and the activity of engineered guide RNA components or their loading into Cas9 can be controlled by base-pairing interactions with intracellular RNAs[30–32]. Spatiotemporal coupling between endogenous signals and CRISPR-Cas DNA targeting has also been described in natural systems. The type III-A CRISPR-Cas systems encode multi-protein effector complexes[3] that naturally depend on co-transcriptional binding of nascent RNAs to license DNA degradation[33–35]. Nascent RNA binding simultaneously localizes the type III-A complex near target DNA and activates its DNase activity[34,35], thereby ensuring that transcribed DNA targets are degraded in *cis* while identical but repressed protospacers may be conditionally tolerated in prokaryotes[36–39]. It is not known whether nascent RNA binding can be exploited to control DNA targeting by other CRISPR-Cas effectors. We investigate this in the present study by engineering Cas9 fusions that can bind to nascent RNA substrates transcribed near a protospacer in *cis*, thus partially emulating the type III-A targeting mechanism and forming the basis for Transcription-associated Cas9 Targeting (TraCT) in eukaryotic cells. subPAM interactions are employed to ensure that Cas9's basal DNA targeting activity in vivo does not mask the transcription-associated activity conferred by nascent RNA binding.

## Results

### Dynamic ligand-controlled rescue of subPAM interactions

Prior to investigating TraCT we first sought to demonstrate that Cas9 targeting can be dynamically controlled without modulating the functionality or expression of its core protein domains and single guide RNA (sgRNA) components. To this end, we reasoned we could engineer Cas9 fusions that conditionally interact with *cis* elements in proximity to protospacers and thereby stimulate targeting when subPAM interactions limit basal activity. Target site editing was assayed using a single-strand annealing[40,41] (SSA) based reporter system[19] wherein protospacer cleavage drives scarless reconstitution of a disrupted *LYS2* open reading frame (ORF) in *Saccharomyces cerevisiae* yeast (Fig. 1a). Throughout the study, this system enabled isogenic testing of various protospacer-proximal *cis* elements inserted between direct repeats without requiring exogenous repair templates. It was previously shown that fusion to strong zinc finger (ZF) DNA binding domains can constitutively increase Cas9 editing at protospacers with subPAMs when a ZF binding site is located nearby[42]. It was furthermore shown, in the same work[42], that ZF-DNA interactions can compensate for suboptimal activity on NGG PAMs when Cas9's PAM-interacting domain[43] (PID) was genetically attenuated with a single substitution known as 'MT3' (Cas9[MT3]). Although ZFs can now be designed to recognize nearly any DNA sequence[44], they do not confer temporal control over Cas9's activity. We therefore initially tested a single *tetO2* DNA element, the target substrate for wild-type (Wt) TetR and reverse TetR (RevTetR) DNA binding proteins that can be dynamically induced with doxycycline (Dox) to dissociate from or bind *tetO2*, respectively[45]. The (Rev)TetR proteins function as homodimers, so we fused tandem monomers to the C-terminus of Cas9 in accordance with a previously described tandem single-chain (sc) design[46], generating Cas9-scTetR and Cas9-scRevTetR fusions. These were constructed from an isogenic scaffold that included N- and C-terminal nuclear localization signal (NLS) tags and also served as our 'native' Cas9 control throughout the work (Supplementary Fig. 1a; Supplementary Note 1). Our

heterologous protospacer insertion contained an NGG PAM that allowed us to compare editing activities for three PID variants that recognize NGG at different strengths: Wt Cas9 (Cas9[Wt]), Cas9[MT3], and SpCas9-NG with multiple PID substitutions[47] (henceforth, 'NG' or Cas9[NG]). As expected, editing with native Cas9 was unaffected by Dox and consistently lower with the MT3 and NG variants compared to Wt (Fig. 1b). In contrast, Cas9-scTetR fusions achieved similarly high levels of editing with all three PID variants in the *absence* of Dox, though editing with MT3 and NG was reduced to background levels when Dox was present (Fig. 1b). This phenotype was qualitatively reversed when the scTetR domain was replaced with either V10 or V16 variants[45] of scRevTetR: in this context, maximal editing by MT3 and NG was achieved in the *presence* of Dox (Fig. 1c). All Dox-controllable phenotypes strictly required the nearby *tetO2* element (Supplementary Fig. 1b).

To confirm that cleavage by Cas9[NG] fusions is stimulated on physiological timescales we set up a liquid induction assay that allowed us to monitor growth while controlling the scRevTetR(V16) domain with Dox. We used modified yeast strains lacking a downstream SSA repeat or other endogenous repair templates that would rescue cells from persistent protospacer cleavage, thus ensuring a prolonged DNA damage response that slows growth upon targeting. Dox-dependent growth reductions required both the scRevTetR domain and the *tetO2* element (Supplementary Figs. 2a–e). Importantly, when compared with non-inducible controls, this Dox-dependent growth reduction was significantly detectable 260 min post-induction (Supplementary Fig. 2f). Assuming our apparent plate reader doubling time of ~90 min (Supplementary Fig. 2e), this indicates that growth was indeed reduced for a substantial portion of the population in under three generations.

### mRNA binding by Cas9 fusions can facilitate TraCT

Having found that Cas9-sc(Rev)TetR fusions allow for dynamic control over DNA targeting activity when combined with subPAM interactions and a nearby *tetO2* element, we next sought to test whether binding DNA-tethered nascent RNA in *cis* could supplant direct DNA binding at *tetO2*. This was first accomplished with well-described MS2 phage parts: RNA hairpins containing either Uracil (MS2[U]) or Cytosine (MS2[C]) at a crucial position in their loop[48–50], and MS2 coat protein (MCP) variants that can bind those sites co-transcriptionally[50–52]. MCP binds to individual MS2 hairpins as a homodimer[49,53,54] so we again fused single-chain tandem monomers, generating Cas9-scMCP (Fig. 1d; Supplementary Fig. 1a). We first tested reporter strains with two Hairpin Templates (HTs) either upstream or downstream of the protospacer (Supplementary Note 2). It was unclear whether the higher-affinity MS2[C] loops[48–50] would be preferred over Wt MS2[U] loops for our purposes, so we used one of each. This yielded a small but significant increase in editing by PID mutants of Cas9-scMCP compared to native Cas9, but only with hairpins placed upstream (Fig. 1e). To probe whether alternative upstream configurations could improve activity we tested single and double MS2[U] loops, or MS2[C] loops. Editing activities were comparable but we obtained lower *p* values with MS2[U] loops (Supplementary Fig. 3a) so proceeded with one or two *MS2[U]* HTs in all subsequent experiments. Cas9[Wt] and Cas9[Wt]-scMCP controls displayed indistinguishable editing in all strains as expected (Supplementary Fig. 3b).

We then tested the phage lambda (λ) N-boxB system[55], which is orthogonal to MS2[56]. Individual boxB RNA hairpins are bound co-transcriptionally by λ N protein monomers[57–60] or a minimal 21-residue tract (N[2–22]) from its N-terminus[61–63]. We fused two N[2–22] peptides (2xN) to the C-terminus of Cas9, generating Cas9-2xN, and assayed editing with one, two, or three *boxB* HTs upstream of the protospacer (Supplementary Note 2). Considerably higher editing by PID mutants was observed with Cas9-2xN than native Cas9, in particular with two or three *boxB* HTs (Fig. 1f). This Cas9-2xN activity was specific for *boxB*

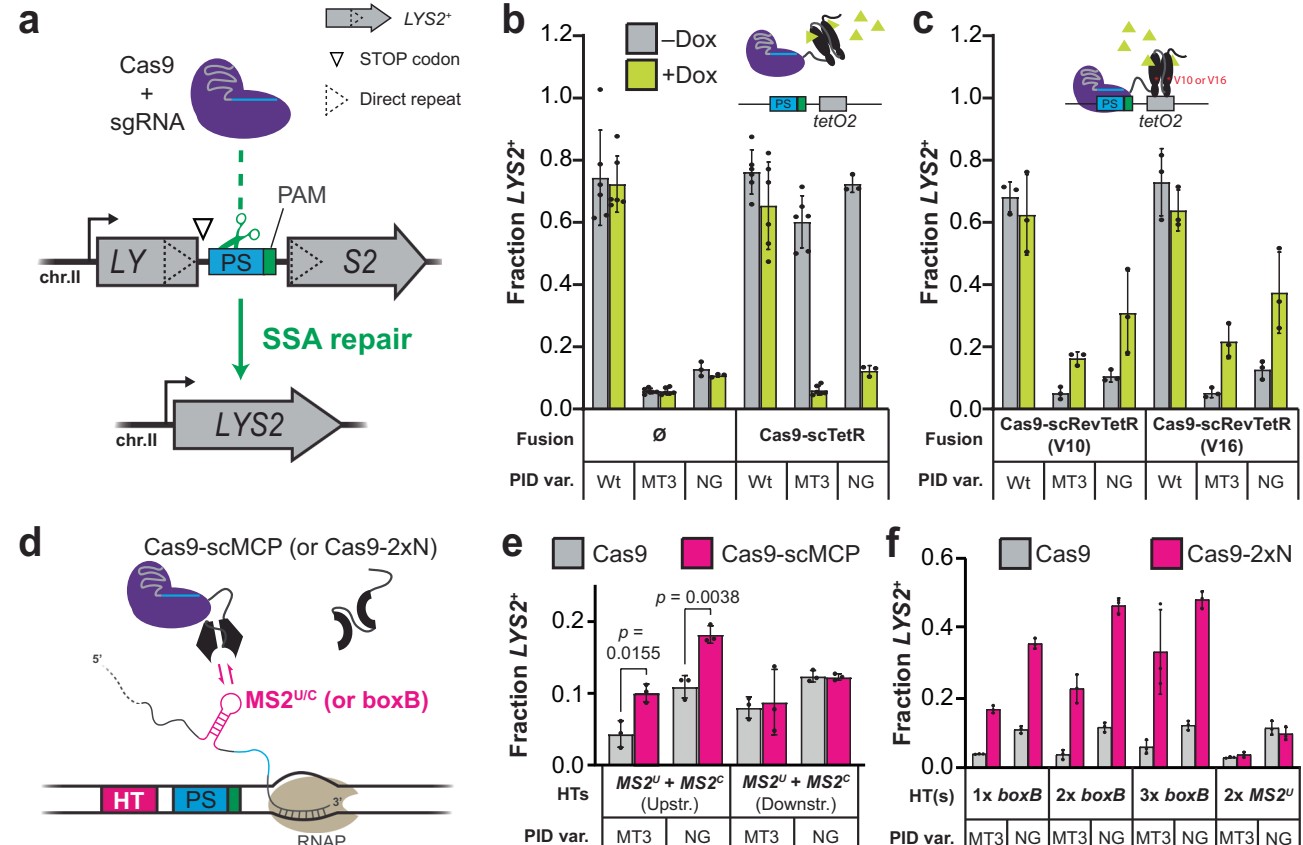

**Fig. 1 | Cas9 activity can be controlled by conditional interactions between auxiliary fusion domains and protospacer-proximal *cis* elements. a** General schematic of the *LYS2* genetic reporter system, engineered with a heterologous protospacer (PS) insertion between 100 bp direct repeats. Co-expression of Cas9 with a targeting sgRNA licenses cleavage and deletion of the repeat-intervening sequences through SSA-based recombination that restores *LYS2*⁺. The native *pLYS2* promoter (bent arrow) and an NGG PAM are used throughout the work, except where stated otherwise. **b** Transformation-associated editing with the Cas9-scTetR fusion or native Cas9 control (Ø) in yeast containing a *tetO2* element 13 bp downstream of the PS. The edited fraction of *LYS2*⁺ transformants (out of total transformants) was measured for Wt, MT3, and NG PID variants using synthetic dropout plates (see "Methods") that lack (−) or contain (+) Dox. Inset schematic illustrates Dox-induced dissociation of the scTetR domain. Error bars, mean ± s.d. ($n = 6$ or $n = 3$, biological replicates). **c** Same as in b but with Cas9-scRevTetR fusions as indicated (V10 or V16). Inset schematic illustrates Dox-induced binding of the scRevTetR domain. **d** Simplified schematic illustration of the Cas9-scMCP fusion interacting with a nascent MS2$^{U/C}$ RNA hairpin, generated from a single Hairpin Template (HT) upstream of the PS. The scMCP domain can be replaced with a 2xN domain that in principle allows binding of two boxB hairpins simultaneously. The nascent transcript is elongated at its 3′ end by an RNA polymerase (RNAP) transcribing left to right. **e** Transformation-associated editing fractions measured for MT3 and NG variants of Cas9-scMCP or native Cas9 in strains containing the indicated HTs. Error bars, mean ± s.d. ($n = 3$, biological replicates). $p$ values were calculated using unpaired, two-tailed $t$ tests with Welch's correction, Gaussian distributions assumed, and no correction for multiple comparisons. **f** Same experimental setup as in e but performed with the 2xN domain in place of scMCP and different HT strains as indicated. Key elements used to create images in this figure were adapted from "Engineered dual selection for directed evolution of SpCas9 PAM specificity." *Nat Commun* 12, 349 (2021) by Goldberg, G. W. et al., under CC BY 4.0. Source data are provided as a Source Data file.

HTs, as no increase was observed with two upstream *MS2*$^U$ HTs. In addition, 2xN fusions partially increased editing with Wt PIDs at suboptimal NAG or NGA PAMs with a single *boxB* HT upstream, though activity at NGG PAMs was independent of 2xN (Supplementary Fig. 3c).

To rule out the possibility that nascent RNA binding nonspecifically increases the activity of interacting Cas9 fusion proteins, e.g., by generally capturing higher concentrations of Cas9 fusions in the nucleus rather than specifically increasing editing near HTs, we tested whether HTs stimulate editing in *trans* using SSA dual-reporter yeast strains. These strains contain a second identical protospacer and PAM insertion, accompanied by zero or two upstream *MS2*$^U$ HTs, that disrupts the constitutively transcribed *CAN1* ORF on a separate chromosome from *LYS2* (Fig. 2a). Once reconstituted through SSA, the *CAN1* locus sensitizes cells to canavanine[64] (Can) and therefore serves as a counterselection marker: only edited cells suffer reduced colony formation efficiency in the presence of Can. Editing at *LYS2* by the NG variant was increased with scMCP fusions only when HTs were present at the *LYS2* locus (Fig. 2b, c). Conversely, whereas scMCP fusions

increased editing by NG at the *CAN1* locus when HTs were present there, they did not increase editing at the *CAN1* locus when HTs were present at *LYS2* (Fig. 2c; Supplementary Fig. 4a). Wt control activities at each locus were not affected by fusion to scMCP or the presence of HTs (Supplementary Figs. 4b–c). Taken together, these results indicate that auxiliary RNA binding domains (RBDs) fused to Cas9 partially compensate for subPAM interactions when their RNA substrates are transcribed from one or more templates near the protospacer in *cis*. Given their co-transcriptional binding capabilities[50–52,57–60], we inferred that these RBDs stimulate editing through a TraCT mechanism.

### TraCT can drive allele-specific editing in yeast

Encouraged by our dual-reporter results, we envisioned applying TraCT in eukaryotic cells to facilitate differential editing at identical target sequences that are transcribed at different levels. To test this potential application and more rigorously challenge the hypothesis that Cas9-RBD fusions function with *cis*-acting HTs via a TraCT mechanism, we constructed a diploid SSA reporter yeast strain for

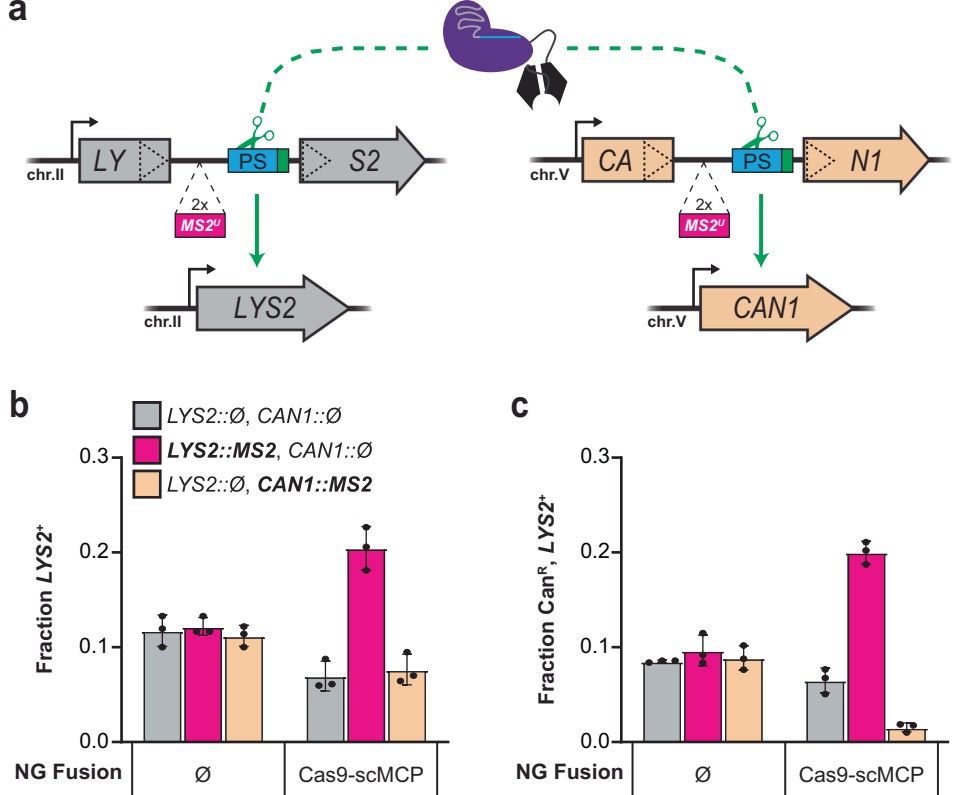

**Fig. 2 | Nascent RNA binding is insufficient to compensate for subPAM interactions in *trans*. a** Schematic summary of the *LYS2* and *CAN1* dual-reporter system, engineered with identical protospacer and PAM insertions between direct repeats at each locus. SSA-based editing can occur via cleavage at either or both loci. Reconstitution of the *LYS2* gene confers lysine prototrophy as in Fig. 1a, whereas reconstitution of the *CAN1* gene confers canavanine sensitivity for counterselection; only cells that were unedited at their *CAN1* locus can form canavanine-resistant (Can[R]) colonies. Strains contained either no HTs, or two upstream *MS2[U]* HTs inserted at one of their loci. The native constitutive promoters (bent arrows) were left unmodified at each locus. **b** Transformation-associated editing fractions measured for Cas9[NG]-scMCP or native Cas9[NG] (Ø) in dual-reporter strains harboring (::MS2) or lacking (::ΔHT) the HT insertions at each locus. Measurements were performed as in Fig. 1e. Error bars, mean ± s.d. (*n* = 3, biological replicates). **c** Editing measurements obtained with the same transformation samples tested in panel (**b**) except using canavanine-supplemented synthetic dropout plates that select only for Can[R], *LYS2*⁺ colonies. Error bars, mean ± s.d. (*n* = 3, biological replicates). Key elements used to create images in this figure were adapted from "Engineered dual selection for directed evolution of SpCas9 PAM specificity." *Nat Commun* **12**, 349 (2021) by Goldberg, G. W. et al., under CC BY 4.0. Source data are provided as a Source Data file.

transcription-dependent *allelic* editing. Our diploid strain was engineered to contain identical target and *MS2[U]* insertions disrupting each of its *LYS2* alleles, but with different promoters controlling their transcription (Fig. 3a). One allele contains a β-Estradiol (β-E) inducible promoter (*pZEVi*) that can achieve high levels of transcription[65,66], while the other contains the native *pLYS2* promoter that is constitutive but weak[67,68]. We first tested haploid yeast strains containing either the constitutive *pLYS2* promoter or the inducible *pZEVi* promoter. In the *pLYS2* strain, editing with Cas9[MT3] or Cas9[MT3]-scMCP was unaffected by β-E as expected (Fig. 3b). In the *pZEVi* strain, however, editing was significantly increased by β-E at 1000 nM relative to 0 and 15 nM, and Cas9[MT3]-scMCP activity at 1000 nM approached that of Wt (Fig. 3b; Supplementary Fig. 5a). Similar trends were obtained when plating diploid transformants after 24 h outgrowth in liquid medium that selects for Cas9(-scMCP) and sgRNA maintenance but lacks β-E (Supplementary Fig. 5b). To definitively determine whether editing by the MT3 variant after outgrowth is preferentially stimulated at the *pZEVi* allele upon induction, we subcultured the diploid transformants in liquid media containing 0, 15, or 1000 nM β-E and then harvested genomic DNA for targeted deep sequencing (see "Methods"). This confirmed that the ratio of unedited *pLYS2:pZEVi* alleles increased significantly in MT3 populations with 1000 nM β-E relative to 15 nM, and to a much greater extent for the scMCP fusion (Fig. 3c). A similarly significant increase in this ratio, though smaller, was also detectable with Cas9[MT3]-scMCP at 15 nM β-E relative to 0 nM, yet it was

undetectable with native Cas9[MT3]. Note that in the absence of β-E we measured a mean ratio slightly *less* than 1:1 (~0.95) with the scMCP fusion, but not with native Cas9 (-1.02), implying that nascent RNA binding at the weakly transcribed *pLYS2* allele was sufficient to license preferential editing relative to the repressed *pZEVi* allele. Collectively, these results establish an engineering strategy that facilitates spatio-temporal coupling between in vivo transcription near particular protospacers and Cas9 targeting activity at those sites.

## TraCT can stimulate editing with exogenous donor DNA

We investigated whether TraCT activity is detectable in non-SSA repair contexts to probe its generalizability. First we designed a haploid *pZEVi-LYS2* yeast reporter strain that lacks the downstream repeat used for SSA, such that perfect reconstitution of its *LYS2* ORF through homology-directed repair requires co-transformation with an exogenous DNA donor fragment (Supplementary Fig. 6a). This strain contains one *boxB* HT upstream of the protospacer for testing with Cas9-2xN fusions (Supplementary Fig. 6a). Constitutive Wt Cas9 cleavage suppresses yeast colony formation until target sequences are genetically disrupted[69,70], and colony formation with Cas9[Wt] or Cas9[Wt]–2xN was increased over 50-fold in our system by supplying a DNA donor fragment that eliminates the target upon repair (Supplementary Fig. 6b). However, unlike Wt neither Cas9[NG] nor Cas9[NG]–2xN suppressed colony formation under any conditions tested; they instead produced only reduced-growth phenotypes with targeting

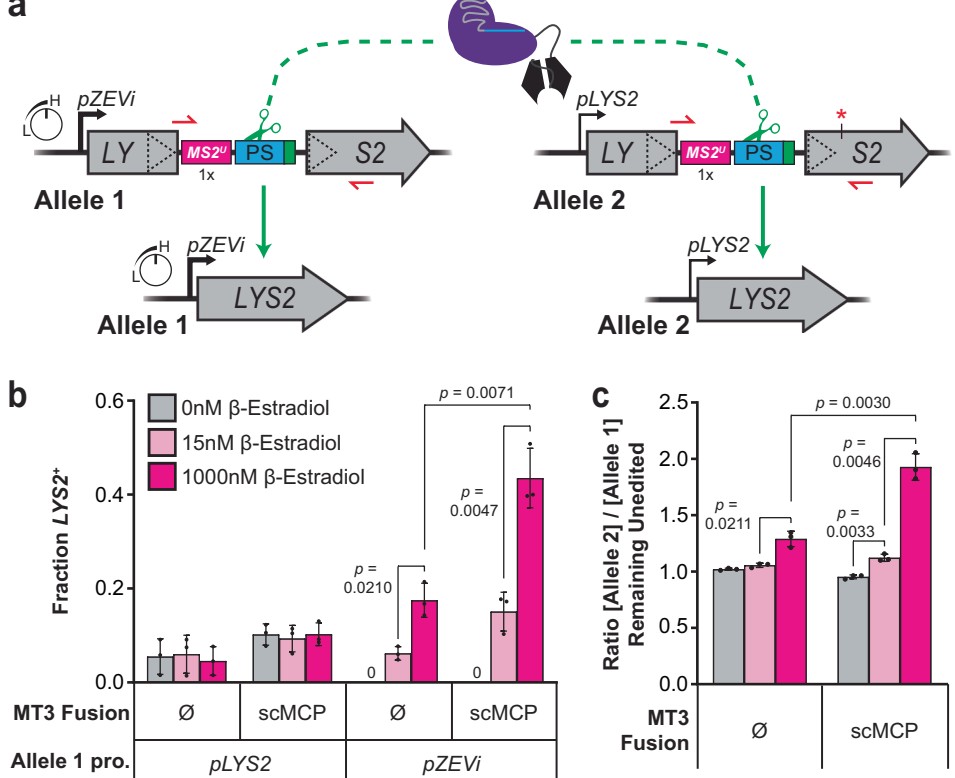

**Fig. 3 | TraCT can be exploited for preferential editing at differentially transcribed alleles with identical target sequences. a** Schematic summary of the diploid SSA reporter system. Each allele differs only in its promoter sequences and a single synonymous point mutation (red asterisk) downstream of identical direct repeat and intervening sequences that disrupt *LYS2* as shown. Allele 1's *pZEVi* promoter can be highly induced with β-Estradiol (β-E) but is repressed in its absence; hence, Allele 1 cannot confer a *LYS2+* phenotype in the absence of β-E, even if reconstituted via SSA. **b** Transformation-associated editing fractions measured for Cas9MT3-scMCP or native Cas9MT3 (Ø) in haploid strains containing only Allele 1 with the *pZEVi* or native *pLYS2* promoter. Measurements were performed as in Fig. 1e except that transformants were also plated in parallel on synthetic dropout plates supplemented with β-E at the indicated concentrations. Error bars, mean ± s.d. (*n* = 3, biological replicates). Zero (0) *LYS2+* colonies of the *pZEVi* strain were detected in the absence of β-E at the time of scoring. *p* values were calculated using

unpaired, two-tailed *t* tests with Welch's correction, Gaussian distributions assumed, and no correction for multiple comparisons. **c** Allelic editing in diploid transformants with Cas9MT3-scMCP or native Cas9MT3, determined by deep sequencing PCR amplicons to distinguish between alleles via the point mutation shown in panel (**a**). Amplicons were generated with a primer pair that specifically amplifies *unedited* alleles (depicted as small red half arrows in panel (**a**)). Data are plotted as the ratio of unedited Allele 2:Allele 1 genotypes measured at each β-E concentration (color coded as in panel (**b**)). Error bars, mean ± s.d. (*n* = 3, biological replicates). *p* values were calculated using unpaired, two-tailed *t* tests with Welch's correction, Gaussian distributions assumed, and no correction for multiple comparisons. Key elements used to create images in this figure were adapted from "Engineered dual selection for directed evolution of SpCas9 PAM specificity." *Nat Commun* **12**, 349 (2021) by Goldberg, G. W. et al., under CC BY 4.0. Source data are provided as a Source Data file.

sgRNAs at 1000 nM β-E (Supplementary Fig. 6b), and their phenotypes were indistinguishable in the absence of donor DNA (Supplementary Fig. 6b; Supplementary Figs. 7a–d; Supplementary Note 3). In parallel, we quantified *LYS2+* fractions by plating immediately (Supplementary Fig. 6c) or by replating after 44 h outgrowth on full plates containing 0, 15, or 1000 nM β-E (Supplementary Figs. 6d–e). This revealed significant increases in donor-dependent editing with Cas9NG–2xN after outgrowth at 0 nM and 15 nM (Supplementary Fig. 6d; Supplementary Note 3).

### Protospacer orientation influences TraCT outcomes

The preceding sections defined several editing conditions in which RBD-mediated interactions can detectably compensate for subPAM activity. Given that subPAM interactions are expected to limit Cas9 recruitment to DNA at the PAM-recognition step, RBDs may license TraCT through a facilitated recruitment mechanism. However, the experiments with *pZEVi* strains also showed that β-E induction can stimulate targeting by native Cas9NG (Supplementary Figs. 6b–e, Supplementary Fig. 7a) or native Cas9MT3 (Fig. 3b, c, Supplementary Fig. 5b) in certain contexts; i.e., when RBDs are absent. This prompted us to further investigate two ways in which transcription might

inherently affect Cas9 editing activity and potentially mask RBD-mediated TraCT.

First, high transcriptional activity has been shown to increase histone eviction[71], and occlusion of PAM sequences within nucleosomes was previously found to impair Cas9 cleavage activity[72]. Therefore, high-level transcription may increase access to naked DNA and thus subvert kinetic or energetic barriers for subPAM interactions. Using MNase-seq and a *pZEVi-LYS2+* haploid yeast strain (see "Methods"), we confirmed that induction with 1000 nM β-E for 7 h reduced median relative nucleosome occupancy at the *LYS2* locus to significantly lower levels than 15 nM (~58% and 84% of the 0 nM median, respectively; Supplementary Figs. 7e–f). Under these conditions, *LYS2* RNA levels were induced more than 100-fold (Supplementary Fig. 7g).

Second, transcribing RNA polymerases (RNAPs) might dislodge native Cas9 from cleaved protospacers in the PAM-downstream orientation employed, thereby facilitating repair outcomes[73]. Relative to the direction of transcription, this corresponds to an sgRNA spacer that anneals to the RNAP's template strand. High transcription can thus ensure frequent Cas9 turnover and rapid processing of cleaved DNA, whereas weak transcription may allow Cas9 to remain bound for longer periods of time and thereby delay processing. When

**a**

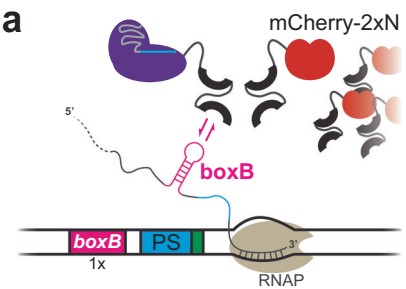

mCherry-2xN

boxB

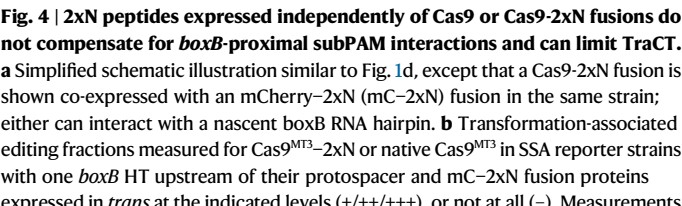

boxB
1x
PS
RNAP

**b**

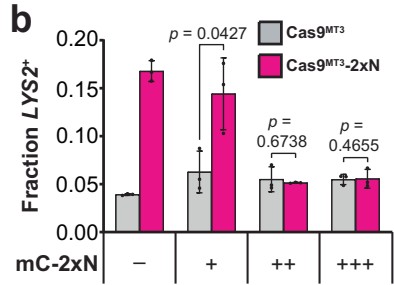

**Fig. 4 | 2xN peptides expressed independently of Cas9 or Cas9-2xN fusions do not compensate for *boxB*-proximal subPAM interactions and can limit TraCT.** **a** Simplified schematic illustration similar to Fig. 1d, except that a Cas9-2xN fusion is shown co-expressed with an mCherry−2xN (mC−2xN) fusion in the same strain; either can interact with a nascent boxB RNA hairpin. **b** Transformation-associated editing fractions measured for Cas9^MT3−2xN or native Cas9^MT3 in SSA reporter strains with one *boxB* HT upstream of their protospacer and mC−2xN fusion proteins expressed in *trans* at the indicated levels (+/++/+++), or not at all (−). Measurements

were performed as in Fig. 1f; data shown for the strain lacking mC−2xN (−) are replotted from Fig. 1f (1x *boxB*, MT3) to facilitate comparisons. Error bars, mean ± s.d. ($n = 3$, biological replicates). *p* values were calculated using unpaired, two-tailed *t* tests with Welch's correction, Gaussian distributions assumed, and no correction for multiple comparisons. Key elements used to create images in this figure were adapted from "Engineered dual selection for directed evolution of SpCas9 PAM specificity." *Nat Commun* **12**, 349 (2021) by Goldberg, G. W. et al., under CC BY 4.0. Source data are provided as a Source Data file.

we tested a protospacer in the inverted (PAM-upstream) orientation with 1 or 3 upstream *boxB* HTs using our original *pLYS2* haploid background (Supplementary Fig. 8a), TraCT was virtually undetectable (Supplementary Fig. 8b). Control experiments with Cas9-scTetR confirmed that the inverted protospacer orientation does not limit MT3 activity if the *tetO2* element is also relocated upstream (Supplementary Figs. 8c–d). Importantly, baseline editing activity with native Cas9 was unaltered by these inversions. In other words, weak transcription from *pLYS2* does not detectably stimulate native Cas9^MT3 or Cas9^NG editing outcomes with one protospacer orientation relative to the other, though it does stimulate Cas9^MT3−2xN or Cas9^NG−2xN via boxB with the PAM-downstream protospacer orientation (Supplementary Note 4).

### TraCT requires a direct Cas9-RBD linkage

An assumption of our proposed model for facilitated recruitment is that direct physical interactions between Cas9 and nascent RNA, as afforded by the covalent linker used in our Cas9-RBD fusions, may be essential for engineered TraCT. To investigate this possibility and further refine our engineering guidelines, we expressed RBDs in *trans* to test whether they suffice to stimulate native Cas9^MT3 when a compatible HT is provided. We tested three different constitutive promoter strengths driving ectopic expression of a nuclear-localized mCherry-2xN (mC-2xN) fusion protein in a *LYS2* SSA reporter strain containing one *boxB* HT upstream of its protospacer insertion (Fig. 4a; Supplementary Figs. 9a–b). Editing with native Cas9^MT3 was not appreciably stimulated at any mC-2xN expression strength (Fig. 4b). Importantly, we also observed a complete abrogation of Cas9^MT3−2xN's TraCT activity when expressing moderate (++) and high (+++) levels of mC-2xN (Fig. 4b) even though expression of mC-2xN had no effect on Wt editing levels (Supplementary Fig. 9c). The inhibitory effect of mC-2xN on Cas9^MT3−2xN likely arises from competitive binding at actively transcribed boxB hairpins. Control experiments were conducted wherein the same three promoters were used to express nuclear-localized TetR proteins in *trans* with MT3 mutants of native Cas9 or Cas9-scTetR targeted to a *tetO2*-adjacent protospacer (Supplementary Figs. 9a, d). In contrast to results with mC-2xN, native Cas9^MT3 activity in the absence of Dox was partially stimulated by *trans* TetR expression when compared with maximal Cas9^MT3-scTetR activity measured without *trans*-expressed TetR (Supplementary Fig. 9e–f; Supplementary Note 5).

To determine whether *boxB* transcription stimulates more Cas9-2xN than native Cas9 recruitment to DNA, we performed chromatin immunoprecipitation (ChIP) experiments that exploit the HA epitope tag present in both proteins (Supplementary Fig. 1). We tested our non-SSA *pZEVi* strain with one *boxB* HT in *LYS2* (Supplementary

Fig. 6a) and a non-targeting sgRNA plasmid to ensure that growth was not significantly perturbed at any β-E concentration. ChIP-qPCR results were highly concordant with data from editing experiments: transcriptional induction with 15 or 1000 nM β-E increased *LYS2* pull-down near *boxB* but not pull-down of *TAF10* DNA from a separate chromosome, and *LYS2* pull-down was significantly higher with Cas9^NG−2xN than Cas9^NG at both induction strengths (Fig. 5a, b). We also performed ChIP-seq for the samples from our first replicate and observed broad increases in pull-down of the *LYS2* locus upon induction, which increased much further with Cas9^NG-2xN in the region beyond *boxB* (Fig. 5c). Although enhancements in *LYS2* pull-down with Cas9^NG-2xN were highest immediately downstream of *boxB* (Fig. 5c, insets), they remained prominent from there down to the terminator region where transcripts dissociate, suggesting that many boxB-containing nascent transcripts are bound throughout this sizeable stretch of the locus. Background pull-down of adjacent genes, or the *TAF10* control region, was virtually identical with Cas9^NG-2xN and native Cas9^NG in the presence of β-E (Supplementary Fig. 10). Therefore, pull-down of *LYS2* was indeed *specifically* enhanced by the boxB interaction. Highly consistent results were observed in experiments where we tested our targeting sgRNA under identical conditions but with dCas9^NG(−2xN) to prevent cleavage-mediated toxicity (Supplementary Fig. 11). Altogether, these results indicate that physical linkages between RBDs and Cas9 license engineered TraCT, likely by increasing Cas9's recruitment near protospacers when cognate HT DNA is transcribed upstream.

### Cas9-RBDs can exploit anchored hairpins

Having confirmed that engineered TraCT functions at least partially through an RBD-facilitated recruitment mechanism, we next examined whether Cas9-RBD fusions can compensate for subPAM interactions by binding RNA hairpins anchored in *cis*, rather than strictly relying on HTs transcribed in *cis*. To test this, we constitutively localized *trans*-expressed boxB hairpins near a protospacer that contains a subPAM but lacks *cis*-acting HTs. This was achieved with a dual sgRNA system for Cas9^Wt-2xN and two closely paired protospacers, in which one contains a suboptimal NGA PAM and the other contains an NGG PAM that can be targeted with partially mismatched spacers which license constitutive binding but not cleavage (Fig. 6a). An array of three boxB hairpins was fused to the 3' end of one sgRNA scaffold (Supplementary Note 2) while the other was left unmodified, and spacers for each target site were then tested in either scaffold. In the first configuration, hairpins are fused to the NGG-targeting sgRNA with spacer 1 (Fig. 6a); in the second, hairpins are fused to the NGA-targeting sgRNA with spacer 2 (Supplementary Fig. 12a). Either configuration could in

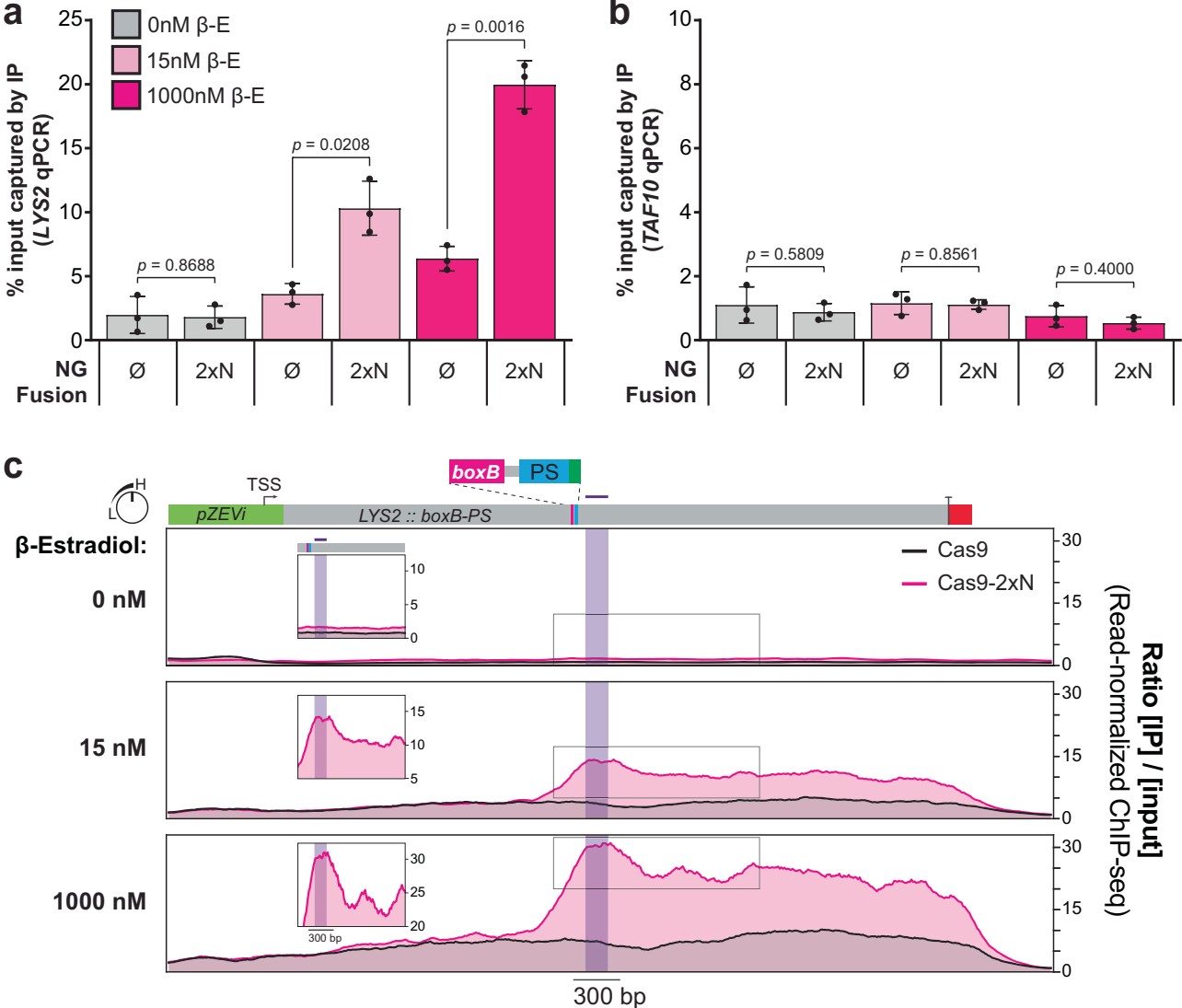

**Fig. 5 | Transcription of *boxB* stimulates Cas9–2xN accumulation at downstream DNA sequences within the transcribed locus. a** ChIP-qPCR for yeast strains carrying a non-targeting sgRNA and Cas9$^{NG}$ (Ø) or Cas9$^{NG}$-2xN (both of which bear a 3xHA tag, see Supplementary Note 1), either untreated (0 nM) or treated (15 and 1000 nM) with β-E for transcriptional induction of the *boxB*-tagged *pZEVi-LYS2* locus. The percentage of chromatin captured by immunoprecipitation (IP) with anti-HA antibody out of total input chromatin was quantified using *LYS2*-specific primers for an amplicon within 250 bp downstream of *boxB*. Error bars, mean ± s.d. (*n* = 3, biological replicates). *p* values were calculated using unpaired, two-tailed *t* tests with Welch's correction, Gaussian distributions assumed, and no correction for multiple comparisons. **b** Background control reactions performed in parallel for all panel (**a**) samples, with qPCR primers amplifying the *TAF10* locus on a separate chromosome that lacks *boxB*. Error bars, mean ± s.d. (*n* = 3, biological replicates). *p* values were calculated using unpaired, two-tailed *t* tests with Welch's correction, Gaussian distributions assumed, and no correction for multiple comparisons. **c** ChIP-seq enrichments across the *boxB*-tagged *pZEVi-LYS2* locus, determined with anti-HA antibody for Cas9$^{NG}$ and Cas9$^{NG}$-2xN at each β-E concentration. Data were generated from a single batch of samples tested in panel (**a**) and plotted as the ratio of IP coverage over input coverage after normalizing each sample for read depth (*n* = 1, biological replicates). The region downstream of *boxB* that was probed by qPCR is delineated with purple shading and a purple line above *LYS2* in the scaled linear map atop the plots. Boxed insets are rescaled to dimensions that emphasize relative differences, or lack thereof, between the maximum values near *boxB* and other local maxima further downstream. TSS transcriptional start site, PS protospacer. Source data are provided as a Source Data file.

principle stimulate Cas9$^{Wt}$-2xN recruitment to the NGA PAM via constitutive interactions at the NGG-adjacent protospacer. Indeed, the 2xN fusion robustly stimulated editing with each sgRNA configuration when 5 mismatches (Mm), but not 20 Mm that completely abrogate binding, were present in spacer 1 (Fig. 6b; Supplementary Fig. 12b; Supplementary Note 6). We also tested modified sgRNA scaffolds with only one or two boxB hairpins and this revealed smaller but still significant increases in editing with Cas9-2xN when spacer 1 contained 5 Mm (Supplementary Figs. 12c–d). These results establish that actively transcribed RNA substrates are not strictly required to facilitate Cas9-RBD targeting at subPAMs, as even *trans*-expressed RNA substrates can facilitate targeting if they are anchored near the protospacer via

Cas9-mediated R-loops. It should be emphasized that the strong NGG PAM we exploited for R-loop formation in this assay is expected to constitutively stimulate targeting at the subPAM, rather than license transcription-associated targeting. However, anchored boxB hairpins are not strictly sufficient to stimulate targeting either, as inversion of the NGA-linked protospacer eliminated Cas9$^{Wt}$–2xN's advantage over native Cas9$^{Wt}$ (Supplementary Figs. 12e–f; Supplementary Note 6).

## TraCT can function in human cells

Finally, we tested whether TraCT can stimulate indel formation through donor-independent non-homologous end-joining (NHEJ) in EGFP reporter lines derived from human U2OS osteosarcoma cells. To

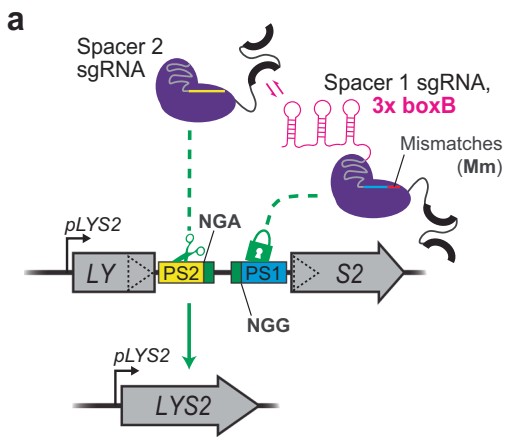

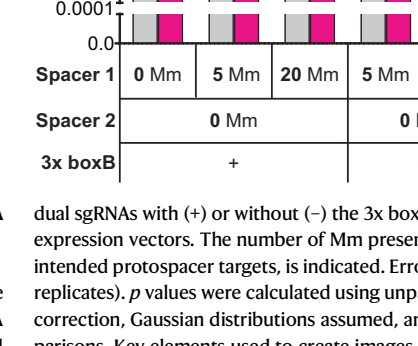

**Fig. 6 | Constitutive R-loop formation with boxB-fused sgRNAs that bind DNA but do not license cleavage can stimulate Cas9$^{Wt}$-2xN targeting at nearby subPAMs. a** Schematic summary of the paired-protospacer SSA reporter system with dual sgRNA programming for Cas9$^{Wt}$-2xN. Three boxB hairpins are fused to the Spacer 1 sgRNA targeting Protospacer 1 (PS1) but not fused to the Spacer 2 sgRNA targeting Protospacer 2 (PS2). Mismatches (Mm) at the 5′ end of Spacer 1 (depicted as the red segment) do not prevent binding but prevent cleavage at PS1. The PAM sequences at each target are denoted 5′ to 3′ irrespective of their orientation. **b** Transformation-associated editing fractions measured for Cas9$^{Wt}$-2xN or native Cas9$^{Wt}$ in the SSA reporter strain summarized in panel (**a**). Various combinations of dual sgRNAs with (+) or without (−) the 3x boxB fusion were delivered via plasmid expression vectors. The number of Mm present in each spacer, relative to their intended protospacer targets, is indicated. Error bars, mean ± s.d. ($n = 3$, biological replicates). $p$ values were calculated using unpaired, two-tailed $t$ tests with Welch's correction, Gaussian distributions assumed, and no correction for multiple comparisons. Key elements used to create images in this figure were adapted from "Engineered dual selection for directed evolution of SpCas9 PAM specificity." *Nat Commun* **12**, 349 (2021) by Goldberg, G. W. et al., under CC BY 4.0. Source data are provided as a Source Data file.

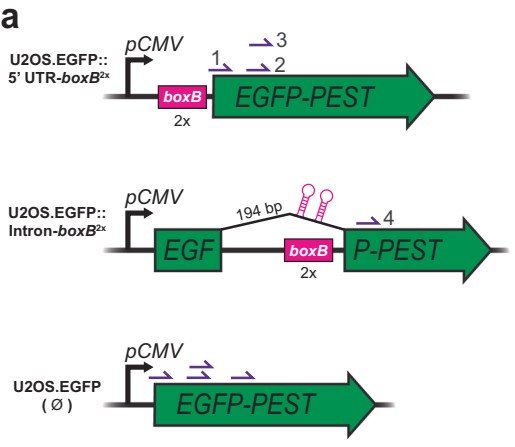

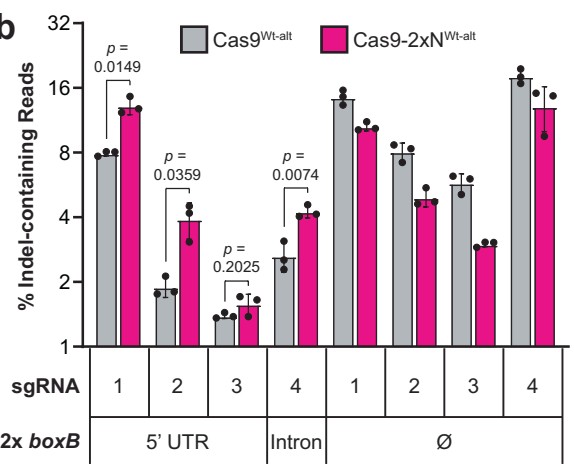

**Fig. 7 | Engineered TraCT can stimulate indel formation in human cells. a** Schematic summary of the non-essential *EGFP-PEST* ORF targeted in human cell culture experiments with the U2OS.EGFP (ΔHT) parent line and its U2OS.EGFP::5′UTR-*boxB*$^{2x}$ or U2OS.EGFP::Intron-*boxB*$^{2x}$(V1.0) derivatives. Relative positions of the four sgRNA targets (purple half arrows with numbering) and two upstream *boxB* HT insertions within the 5′ UTR or 194 bp synthetic intron are indicated. Transcription is driven by the strong constitutive *pCMV* promoter in all cell lines. **b** Deep sequencing of protospacer-spanning amplicons to quantify the percentage of reads that contain indels, inferred to result from donor-independent NHEJ. gDNA templates were extracted 48 h after co-delivering one of four targeting sgRNA plasmids (1 through 4) and the Cas9-2xN$^{Wt-alt}$ fusion or control Cas9$^{Wt-alt}$ plasmid into the indicated cell lines. The PAMs CGAGG, CGAGC, GAGCT, and TGACC license targeting for sgRNAs 1 through 4, respectively. Error bars, mean ± s.d. ($n = 3$, biological replicates). $p$ values were calculated using unpaired, two-tailed $t$ tests with Welch's correction, Gaussian distributions assumed, and no correction for multiple comparisons. Source data are provided as a Source Data file.

this end, two *boxB* HTs were inserted into either the 5′ UTR of the *EGFP* marker cassette or an intron within the *EGFP* ORF, and downstream protospacers with NGA or NAG subPAMs were each targeted with one of four sgRNAs (Fig. 7a). When the U2OS.EGFP control line lacking *boxB* HTs was co-transfected with transient expression plasmids for Cas9 and sgRNA, we measured lower indel formation with an alternative Wt Cas9-2xN fusion (Cas9-2xN$^{Wt-alt}$) relative to the isogenic native Cas9$^{Wt-alt}$ architecture (Fig. 7b; Supplementary Fig. 1c; Supplementary Note 1). Despite this apparent impairment in the absence of *boxB*, indel formation by Cas9-2xN$^{Wt-alt}$ was significantly higher (1.61- to 2.07-fold) relative to Cas9$^{Wt-alt}$ with three out of four sgRNAs in the

*boxB*-containing lines (Fig. 7b). These results provide a proof-of-principle that TraCT can function in human cells.

## Discussion
We report here that spatiotemporal control over Cas9 targeting in eukaryotic cells can be achieved using engineered Cas9 fusion proteins that conditionally interact with specific DNA or RNA substrates in proximity to protospacers. Unlike other engineered strategies for inducible Cas9 targeting that modulate the expression or activity of Cas9's core domains and sgRNA components[21–32], our approach offers conditional targeting activity only at protospacers with subPAM

interactions and suitably positioned *cis* elements. Hence, our Wt PID fusions were fully active and targeted strong PAMs constitutively under all conditions tested. This implies that conditional *and* constitutive targeting could be simultaneously achieved at two unlinked protospacer sites if one has a subPAM and the other a strong PAM.

Central to this work, fusion of Cas9 to our scMCP or 2xN RBDs compensated for subPAM interactions through a TraCT mechanism that exploits *cis*-acting HTs. This activity is inherently conditional because the RNA hairpin substrates for these RBDs are only transiently tethered near their HT DNA during active transcription in *cis*. Transcription of HTs in eukaryotic cells can thus serve as an intracellular physiological signal to control Cas9 targeting spatiotemporally, analogous to the role of protospacer transcription in DNA targeting by type III systems in prokaryotes[33–39]. TraCT activity scaled with the level of HT transcription and was thereby advantageous when applied in yeast for partial discrimination between identical allelic sequences that are differentially transcribed; i.e., TraCT facilitated preferential editing at the more highly transcribed of two target alleles in a diploid yeast strain (Fig. 3). Given also the suitable effect sizes we measured with yeast SSA reporters throughout this study, it is expected that our system will be generally applicable in future work as a two-hybrid platform for testing whether other putative or known protein-RNA interactions can function co-transcriptionally. We anticipate that TraCT could be further applied in future studies as a genetic surveillance or lineage tracking system in animal cell populations with heterogeneously transcribed DNA targets[74–78]. For example, stable expression of RBD fusions and sgRNAs in differentiating cells could selectively enrich for editing in subpopulations that upregulate transcription at HT-linked protospacers. However, future improvements in higher eukaryotes may be needed before TraCT can be implemented for such applications, considering that our proof-of-principle indel experiments in human cells produced only small effect sizes. We moreover caution that the utility of existing Cas9-RBD fusions for TraCT may prove limited outside of genetically engineered systems, owing to their reliance on heterologous *cis*-acting HTs, and pre-insertion of HTs may be less efficient in many cell lines relative to our U2OS derivative's efficiency ( ~ 23% for insertion of *boxB*[2x] in the 5' UTR; see "Methods"). Although our double *boxB* insertion within the 5' UTR of *EGFP* did not reduce mean cellular fluorescence (Supplementary Fig. 13), secondary structures inserted within eukaryotic 5' UTRs have been found to reduce expression in certain contexts[79,80], highlighting another potential engineering constraint that could limit TraCT's implementation. Notwithstanding, natural eukaryotic systems also contain proteins capable of binding nascent RNA, such as splicing factors that are co-transcriptionally localized near the DNA templates of introns, splice junctions, or their downstream exons[81–83]. We therefore speculate that fusion of Cas9 to splicing factors could facilitate TraCT at such native sites. It might also be possible to program TraCT virtually anywhere in the genome by targeting pre-mRNA with antisense RNAs fused to boxB hairpins[84,85] for Cas9−2xN recruitment, or, fused directly to Cas9's sgRNA scaffold. Alternatively, fusion of Cas9 to an RNA-targeting (d)Cas13 effector protein[86–88] from type VI CRISPR-Cas systems[3] might prove effective. Whether (d)Cas13 complexes are capable of binding target RNAs co-transcriptionally remains to be elucidated, however.

In light of prior work and findings herein, several lines of evidence offer insight into the molecular details underlying engineered TraCT in eukaryotic cells. Nucleosomal DNA is thought to principally impede Cas9 targeting at the PAM-recognition step[72,89], and DNA binding sites for proteins that displace nucleosomes in vivo have been shown to locally stimulate targeting[90]. Active transcription was also confirmed to reduce nucleosome occupancy in our system (Supplementary Figs. 7e−f), though it need not evict histones from DNA entirely[91]. Hence, we speculate that recruitment of Cas9-RBD fusions to transcribing RNAPs, via tethered nascent RNA, provides temporal windows

during which Cas9 can rapidly interrogate naked PAM DNA in *cis*. Concomitant RNA tethering and nucleosome displacement would thus transiently overcome the kinetic and/or energetic limitations of sub-PAM interactions in eukaryotic cells. In other words, TraCT might function through a dynamic mechanism that relies on concomitant Cas9 recruitment and exposure of target DNA by RNAPs. This two-part model (Supplementary Fig. 14) could explain why upstream HTs offered a pronounced advantage over downstream HTs; downstream HTs will yield RNA hairpin substrates only after an RNAP has transcribed beyond the target, at which point target DNA may once again be occluded within nucleosomes[91]. Supporting this model, accumulation of Cas9[NG]−2xN peaked within 300 bp downstream of *boxB* in our yeast ChIP-seq experiments (Fig. 5c, insets), likely because short RNA tethering lengths maximize local concentrations at exposed DNA near the transcription bubble. Beyond that peak, accumulation remained considerably higher for Cas9[NG]−2xN than native Cas9[NG] but plateaued at a reduced level in the downstream *LYS2* locus, so we infer that tethering Cas9[NG]-2xN at longer lengths may still increase DNA contacts but to a lesser extent that results in fewer crosslinks. Various lengths have been tested in prior studies that examine how boxB-mediated tethering can facilitate Wt λ N's functional interactions with elongating *E. coli* RNAPs. For instance, using a 184 nt length, one study confirmed that interactions with in vitro transcribing RNAPs are reduced when boxB is prematurely cleaved off the nascent RNA via treatment with a precise RNase[59]. It remains to be determined whether TraCT can function with HTs 150-300 bp upstream of the protospacer or further, as all HTs used in this work were less than 150 bp away. In contrast with downstream HTs that largely failed to license TraCT, hairpins constitutively anchored downstream using our paired-protospacer strategy in yeast robustly stimulated editing by Cas9-RBDs (Fig. 6), beyond the proxy-CRISPR effect[92] proposed to result from nucleosome displacement alone. This qualitatively mirrored our results comparing Cas9[MT3]-scTetR to native Cas9[MT3] with TetR in yeast (Supplementary Fig. 9e), and ostensibly supports our two-part TraCT model that jointly considers recruitment and nucleosome displacement effects. Interactions with static *tetO2* elements or NGG-adjacent mismatched protospacers licensed an advantage over native Cas9 in yeast only when positioned on the PAM side of the cleavable protospacer, however (Supplementary Figs. 8c−d; Fig. 6; Supplementary Figs. 12e−f). This might reflect Cas9's reported requirement for a sequence-independent 'post-PAM' interaction at PAM-distal DNA[93], which, should be nucleosome-depleted in these configurations. If we furthermore surmise that transcription dynamically exposes DNA for the post-PAM interaction, the relative kinetics of this step should also be considered when interpreting our result that TraCT was influenced by protospacer orientation at a weakly transcribed yeast locus (Supplementary Figs. 8a−b). Additional experiments will be required to dissect these mechanistic points and investigate whether transcription-associated DNA targeting can be achieved with other CRISPR-Cas effectors in eukaryotic cells.

## Methods

### Microbial strains and growth conditions

Cultivation of *Escherichia coli* strains XL1-Blue or TOP10 was carried out as previously described[19] for petri dish plates or 5 mL liquid cultures. Media were supplemented with carbenicillin (100 μg/mL) and/or kanamycin (50 μg/mL) for plasmid cloning and maintenance. When chemically competent or electrocompetent XL1-Blue cells were prepared for downstream cloning, media was supplemented with tetracycline (15 μg/mL) alone.

The *S. cerevisiae* yeast strains used in this work were all derived from BY4741 or BY4731 haploids[94]; their genotypes are provided in Supplementary Data 2. Except where noted otherwise, all liquid yeast culture and competent cell preparation was performed sterilely with YEPD broth or synthetic complete (SC) dropout broths lacking one or

more amino acid (SC−) in 18 × 150 (for 5 mL volumes) or 25 × 150 mm (for 20 mL volumes) disposable round-bottom glass tubes as described previously[19]. This included our adaption of the high-efficiency lithium acetate transformation protocol[95]. Briefly, 5 mL YEPD or SC starter cultures were first inoculated from single colonies and grown for 24 h overnight. After subculturing overnight starter cultures by diluting to ~0.15 Attenuance ($D_{600\ nm}$) in 20 mL of the same broth (1:100 for YEPD or 1:70 for SC) and allowing 4.5 h (YEPD) or 5.25 h (SC) growth on a roller drum at 30 °C ( ~ 0.6 $D_{600\ nm}$), cells were harvested by centrifugation at 2103 × g for 3 min and washed once with resuspension in an equal volume of sterile double-distilled water (ddH$_2$O). After washing a second time with resuspension in an equal volume of filter-sterilized lithium acetate (0.1 M) and re-pelleting, cells were transferred to 1.5 mL tubes in their residual supernatant, centrifuged again, and resuspended to a final volume of 200 µl in 0.1 M Lithium Acetate. Each 20 mL subculture yielded four transformation reaction aliquots; when scaling up for more transformations, additional subcultures were grown in parallel and cells were pooled in 1.5 mL tubes at this step. A modified transformation mix (TRAFO mix)[95] was prepared fresh in parallel, and 50 µl of cells were combined with 300 µl of TRAFO mix in 1.5 mL tubes for each transformation reaction. DNA ( ≤ 20 µl in ddH$_2$O or buffer EB from Qiagen) was then added directly to this mixture, and tubes were homogenized by vortexing before transferring to a roller drum at 30 °C. After rotating for 30 min, 36 µl pure DMSO was added to each tube, and tubes were re-homogenized by vortexing and transferred to a 42 °C heat block. After incubating for 15 min, cells were pelleted by centrifugation and resuspended in filter-sterilized 5 mM CaCl$_2$ (400 µl, unless noted otherwise) and then incubated at room temperature (r.t.) for 10 min. After 10 min at r.t., resuspended cells were spread directly on solid media (150-200 µl for 100 × 15 mm plates) using sterile 4 mm glass beads (Fisher), or spotted as droplets (5 µl on YEPD or 10 µl on SC) from serial dilutions prepared with sterile ddH$_2$O. Our fresh TRAFO mixes were homogenized from 240 µl PEG 3350 (44%), 36 µl 1 M Lithium Acetate, and 25 µl Promega Herring Sperm DNA (10 mg/mL) per reaction by vortexing vigorously in a 50 mL sterile falcon tube. Immediately before preparing a TRAFO mix, pre-aliquoted Herring Sperm DNA was thawed and resuspended by vortexing vigorously, boiled on a heat block at 95 degrees for 5 min, and then chilled on ice for 5 min. To select for transformation-associated integration of DNA fragments linked to a nourseothricin-resistance (Nat$^R$) resistance marker, cells were plated on solid media supplemented with clonNAT (100 ug/mL) after recovering half the transformant suspension for 16-24 h in the same clonNAT-free liquid media that was used to prepare their parental competent cells. ClonNAT was generally maintained during Nat$^R$ strain re-streaks but not included in the media during other assay procedures. For all experimental yeast assays in this work, biological replicates were initiated by launching liquid starter cultures from fresh single colonies grown on SC dropout plates at 30 °C for 48 h after streaking. Except where noted otherwise, SC media formulations (Sunrise Science) with leucine and/or uracil dropped out were used throughout the work to select for maintenance of the Cas9 and/or sgRNA episomes, respectively. Slight colony size reductions were observed when selecting for strains that express Cas9$^{Wt}$-scMCP with or without a sgRNA (Supplementary Fig. 5c), suggesting low-level toxicity from expression.

## DNA preparation and cloning

All circular plasmid DNA used in this work was purified from saturated 5 mL liquid cultures of *E. coli* (grown for 16 h at 37 °C as described previously[19]) using Qiagen miniprep reagents. Centrifugation at 4000 × g for 6 min was used to pellet the cultures before miniprepping. Plasmid identifiers and descriptions are provided in Supplementary Data 3. All plasmid-derived linear DNA fragments used for cloning, yeast strain construction, or transformation-associated editing experiments were isolated from miniprepped DNA via restriction

digest and gel purification. PCR amplicons used for cloning, CRISPR-assisted knock-in, or Illumina sequencing were also gel-purified. For cloning and CRISPR-assisted knock-ins, amplicons were generally obtained using high-fidelity Q5 Hot Start (2X Master Mix) or Phusion polymerases from NEB; otherwise, GoTaq Green (Promega) was used for fragments up to ~ 2 kb that could not be obtained with Q5 or Phusion. GoTaq was also used for all PCR amplification from genomic DNA (gDNA) templates, including for yeast colony PCRs performed as described previously[19] by templating with 2 µl of supernatant from colonies that were resuspended in 40 µl NaOH (20 mM) and boiled 7 min. Cycling conditions recommended by the manufacturer were usually followed, except that 90 s of annealing was used during each cycle when amplifying from gDNA with GoTaq. All Wt, MT3, and NG PID fragments used for assembly with gel-purified acceptor vector backbone fragments in transformation-associated editing experiments were amplified from plasmid templates using Q5 or Phusion with oligos oGG1019/oGG1052 (Supplementary Data 4) and then gel-purified. Gel purifications were typically carried out with disposable x-tracta tools (Sigma-Aldrich) and MinElute kits (Qiagen). In addition to PCR- or plasmid-derived fragments, commercially synthesized DNA fragments (e.g., gBlocks from IDT) were also used directly in cloning procedures.

The majority of plasmid cloning in this work was carried out with Gibson[96] or NEBuilder HiFi (NEB) master mixes for isothermal assembly of linear DNA fragments at 50 °C for 1 h. In some cases these assemblies included a single-stranded DNA bridging oligo for introducing small inserts, such as sgRNA spacers. Otherwise, cloning was achieved via one-pot restriction-ligation reactions with type IIS restriction enzymes (NEB or Thermo Scientific) and T4 DNA ligase (NEB). These 20 µl reactions included circular plasmid DNA (destination vector) and annealed oligos with appropriate overhangs for ligation, 1 µl type IIS restriction enzyme, 1 µl T4 ligase, and buffers (CutSmart and T4 ligase buffers, each at 1X final concentrations); they were incubated at 37 °C for 4-24 h. In most cases XL1-Blue or TOP10 chemically competent cells were used for cloning and before plating were recovered into 300 µl SOC broth for 1 h on a roller drum at 30 or 37 °C after the 45 s heat shock at 42 °C. If these did not yield clones initially, electrocompetent XL1-Blue cells were occasionally used in accordance with dialysis and electroporation procedures described previously[19]. Following electroporation, transformants were recovered into 10 mL SOC broth for 1 h on a roller drum at 30 or 37 °C before plating. At a minimum, all cloned sequences derived from PCR amplicons or commercially synthesized DNA and their adjacent backbone-derived sequences at recombinant junctions were verified by Sanger sequencing. In some cases, entire plasmids were sequenced with long-read sequencing by Plasmidsaurus. The expected sequences of all plasmids used in this work are provided in Supplementary Data 3.

All Cas9 acceptor vectors were cloned in *E. coli* and served as the source of backbone fragments for assembly with PID inserts in yeast. These were based off the pGG211, pGG431, and pGG442 acceptor vectors[19], which were themselves derived from a minimal Cas9$^{Wt}$-SV40 expression plasmid (pNA0306) described previously[97]. The kanamycin-resistance (Kan$^R$) cassette that replaces PIDs in all acceptor vectors is flanked by restriction sites for backbone isolation and ensures that any uncut or singly cut plasmid carryover does not contribute functional Cas9 activity upon transformation. All backbone fragments were isolated by restriction digest and gel purification; NheI and NotI were used for pGG574, pGG547, pGG548, or pGG618 plasmids (Dox induction experiments) while NheI and PspOMI were used for pGG532, pGG500, or pGG573 plasmids (TraCT experiments). When functional Cas9 plasmids were needed for strain construction they were pre-assembled with PID inserts and cloned in *E. coli* before delivery into yeast. Expression of sgRNAs in yeast was achieved with derivatives of the single sgRNA (pNA0304) or double sgRNA (pNA0308) plasmids described previously[97]. The pGG197 single sgRNA plasmid was used for the majority of targeting assays in this work and

its spacer matches a heterologous *EGFP* protospacer (which served as Protospacer 1 in paired-protospacer experiments) that was employed previously[19]. The Protospacer 2 sequence used in paired-protospacer experiments is also a heterologous *EGFP* protospacer which was targeted by the 'NAG 3' spacer employed previously[18], but its PAM was altered in this work (CAGCTTGC→CGAAGTAA). All derivatives of pNA0308 also included a *SUP4* terminator insertion between the second sgRNA scaffold and the *RPR1* terminator. Constitutive expression of Z₃EV in yeast, the β-E-inducible activator protein for our *pZEVi* promoter, was achieved with an *ACT1* promoter and *RPS5* terminator cassette stably integrated at the *YKL162C-A* locus. This cassette was cloned along with a Nat^R marker and homology arms in a NotI fragment of pGG613, as in pMS101, but without extraneous sequences between the Nat^R marker and downstream homology arm. The pMS101 plasmid includes previously described parts[65] but was redesigned with a pAV10.KN backbone for integration at the *YKL162C-A* locus as described previously[98]. The *pZEVi* promoter used in this work is based off the P4 promoter[66], a *GAL10-1* hybrid promoter with binding sites for Z₃EV. Constitutive expression cassettes for nuclear-localized TetR and mCherry-2xN proteins were similarly cloned in *E. coli* plasmids and then isolated on NotI fragments for stable integration at the *YKL162C-A* locus in yeast. These plasmids were built off the pSIB619 architecture previously used for constitutive expression of the cam-TA protein[99] with a *TDH1* promoter and *STR1* terminator (modified to lack BsmBI restriction sites). The *TDH1* promoter was replaced with *ADH1* or *TDH3* promoters to obtain higher constitutive expression strengths in a subset of the pSIB619 derivatives. The Cas9^Wt-alt (pAG5) and Cas9-2xN^Wt-alt (pGG768) plasmids used for transient expression in human cells were cloned in *E. coli* and derived from the pGG439 plasmid[19], which was itself derived from the MSP469 plasmid[18]. Transient expression of sgRNAs in human cells was achieved with derivatives of the BPK1520 single sgRNA plasmid[18], also cloned in *E. coli*.

**Construction of yeast strains**
All haploid yeast strains constructed in this work are described within Supplementary Data 2 and were generated using one of the following three transformation schemes. (1) To introduce circular episomes carrying selection markers that confer prototrophy, plasmid DNA (~30-60 ng) was delivered directly through transformation and cells were plated with appropriate selection. (2) To modify the native *LYS2* or *CAN1* loci (including *LYS2's* promoter region) we used a CRISPR-assisted knock-in method essentially as described previously[19]. Sequences of the sgRNA plasmids used for knock-in at the *LYS2* ORF (pGG202), *CAN1* ORF (pGG203), or *LYS2* promoter (pGG578) are provided in Supplementary Data 3. Sequences of the linear double-stranded DNA donor fragments used for knock-in, including their homology arms, are listed in Supplementary Data 5. When patching colonies for initial phenotypic screening in this work, SC−Arg−Leu−Lys−Ura plates were used in place of SC−Lys, and SC−Arg−Leu−Lys−Ura+Can plates were used in place of SC−Arg+Can plates. Furthermore, during *pZEVi* strain isolations, 1000 nM β-E was included in the SC−Arg−Leu−Lys−Ura plates to turn on the *pZEVi* promoter and facilitate *LYS2*⁺ colony screening. CRISPR-assisted edits were validated by Sanger sequencing of gDNA-templated PCR amplicons that covered, at a minimum, all recombinant sequences and the flanking homology arms used for knock-in. In some cases, intermediate strains with genotypes that differ from BY4741's were used as parent strains for a second round of CRISPR-assisted knock-in, as outlined in Supplementary Data 2. Before delivering different sgRNA and/or Cas9 plasmids into CRISPR-edited strains for downstream experiments or for secondary rounds of CRISPR-assisted editing, the original sgRNA and/or Cas9 episomes used for knock-in were cured via replica plate screening as described previously[19]. (3) To integrate DNA into the *YKL162C-A* locus[98], linear DNA fragments

(~200-300 ng) bearing a Nat^R marker and sufficient homology arm lengths to select for integrative recombination were directly transformed without CRISPR co-selection; cells were then recovered for 24 h in liquid YEPD or SC dropout media to select for plasmids before plating on solid media of the same type but with clonNAT present. These linear DNA fragments were excised from circular *E. coli* plasmids listed in Supplementary Data 3 via restriction digest with NotI, followed by gel purification. Integration of the pGG613 fragment in yGWG307, yGWG311, and yGWG312 lineages or the pGG658 fragment in yGWG315 was further validated by colony PCR and Sanger sequencing of regions spanning the homology arm junctions in *YKL162C-A*.

The diploid SSA reporter strain yGWG422 was mated from yGWG312 and yGWG381 haploids (Supplementary Data 2) on solid media. In brief, single colonies of each strain were streaked from opposite sides of a YEPD plate into a central overlapping region where mating could occur. After 7 h growth at 30 °C, this plate was replica-plated onto an SC−Ura−His plate to select for segregants with prototrophies unique to each parent strain (*URA3*+ from pGG197 in yGWG312 and *HIS3*+ from yGWG381's genome). A diploid clone was isolated from the segregant pool by re-streaking onto an SC−Ura−His plate; its SSA target region was amplified by colony PCR and Sanger sequencing confirmed the expected heterozygosity (~1:1 allelic ratio) at the position of the point mutation.

**Quantitative transformation-associated editing assays in yeast**
Editing of haploid SSA reporter strains with a single sgRNA was performed essentially as described previously for clonal selection experiments in yeast[19]. In brief, haploid reporter strains were co-transformed via the lithium acetate procedure described above using 150 ng linearized backbone DNA and ~41 ng PID insert DNA (~1:3 molar ratio). Transformant populations resuspended in CaCl₂ were then serially diluted and spotted in parallel on different SC dropout plates, including SC−Leu−Ura plates to select for all viable transformants maintaining both the Cas9 and sgRNA episomes, as well as SC−Arg−Leu−Lys−Ura plates that select for transformants with both episomes and a reconstituted *LYS2* marker. Where applicable, additional spotting on SC−Arg−Leu−Lys−Ura plates supplemented with Dox, Can, or β-E was performed in parallel using the same serial dilutions. Plates were incubated for 47 h at 30 °C (or 72 h for Can-supplemented plates) before immediately photographing and then quantifying colony forming units (CFUs) by eye. After initial scoring, plates were stored at benchtop for up to 16 h (or 24 h for Can-supplemented plates) and counted once more; the higher of two CFU counts was used in determining each *LYS2*⁺ fraction, calculated as the ratio of *LEU2*⁺, *LYS2*⁺, *URA3*⁺ (and Can^R, where applicable) CFUs over *LEU2*⁺, *URA3*⁺ CFUs obtained on SC−Leu−Ura plates. Typically, each Cas9 PID variant fusion protein replicate was tested in parallel with a corresponding native Cas9 PID variant control replicate performed on the same day. The same procedures were followed for editing of paired-protospacer reporter strains with dual sgRNAs except that recipient strains harbored pre-assembled native Cas9^Wt or Cas9^Wt−2xN episomes instead of the sgRNA episome; competent cells were therefore prepared in SC−Leu rather than SC−Ura media and were transformed with circular dual sgRNA plasmid DNA (2.5 μg) instead of linear fragments.

For donor-dependent editing of haploid yeast, all recipient strains harbored pre-assembled native Cas9^Wt/NG or Cas9^Wt/NG-2xN plasmids. These strains were transformed with 75 ng of the targeting sgRNA plasmid (pGG197) alone or co-transformed with the targeting or non-targeting (pGG202) sgRNA plasmids (75 ng) and 120 bp commercially synthesized double-stranded DNA donor fragment (500 ng). Transformants were resuspended in 700 μl CaCl₂ prior to initial platings, performed by spreading aliquots (154 μl per full plate) from the CaCl₂ suspension on SC−Leu−Ura media with or without β-E at 15 nM or

1000 nM. These plates were grown for 44 h at 30 °C before scrape-homogenizing for serial dilution and replating to determine their *LYS2*[+] fractions with SC−Arg−Leu−Lys−Ura plates that contained β-E (1000 nM). Replates were incubated 38.5 h at 30 °C before immediately photographing and quantifying CFUs to determine *LYS2*[+] fractions as described above, except only 12 h were allowed at benchtop before recounting. In parallel with initial platings, aliquots were withdrawn from the remaining CaCl$_2$ suspensions for serial dilution and spotting on the same three solid media conditions. For strains with Cas9$^{NG}$ or Cas9$^{NG}$-2xN, the same serial dilutions were also spotted on SC−Arg−Leu−Lys−Ura plates supplemented with β-E at 15 nM or 1000 nM to quantify *LYS2*[+] fractions obtainable immediately after transformation.

### Plate reader growth assays

To obtain 1 mL SC−Leu−Ura liquid starter cultures for each biological replicate, single colonies isolated by streaking on a single plate were vigorously inoculated and cultured in parallel on the same day. These were grown with continuous shaking (800 RPM) for 22 h at 30 °C in 96-well V-bottom assay blocks (Corning, 2 mL capacity) that were pre-autoclaved (dry) with a 4 mm glass bead in each well. Starter cultures were then subcultured 1:100 into 96-well SensoPlate microplates with a clear, flat bottom (Greiner Bio-One) containing 200 μl of SC−Leu−Ura media in each well, and grown for 32 h at 30 °C with their lids on and continuous shaking in a Cytation 5 microplate reader (BioTek). High-resolution growth curves were generated from Attenuance ($D_{600\ nm}$) measurements taken every 10 min. For Dox induction experiments, microplates were briefly withdrawn after 5 h of growth and cultures were supplemented with Dox (10 μM final concentration) or solvent alone by pipetting directly into the wells, then returned to the Cytation 5 for an additional 27 h of growth. Lids were kept warm in a 30 °C incubator during the withdrawal step. For β-E induction experiments, media containing β-E was supplemented at 15 nM or 1000 nM before aliquoting into microplate wells and was thus present throughout the 32 h.

### Analysis of high-resolution growth curves from plate reader assays

All growth curves were analyzed using custom python scripts. For each curve, $D_{600\ nm}$ measurements were corrected for background by subtracting their minimum measurements from each value, and then doubling times ($t_d$) were calculated by fitting a logistic equation to each curve[100]:

$$N_t = \frac{K}{1 + \left(\frac{K - N_0}{N_0}\right) e^{-rt}} \tag{1}$$

$$N_t = \text{Attenuance} \left(D_{600 nm}\right) at\ time\ t$$

$$t_d = \frac{\ln 2}{r} \tag{2}$$

To calculate time-resolved differences in growth, $D_{600\ nm}$ measurements from +Dox cultures were subtracted from −Dox cultures at each time point. Significance was calculated at each time point by comparing between pooled control samples ($n = 18$) and *tetO2*+ , Cas9$^{NG}$-scRevTetR (V16) samples ($n = 6$). Before calculating significance, the median differences calculated for each pair of cultures over their first 5 h (during which neither culture contains Dox) were assumed due to background variation and subtracted from the differences at each timepoint. Difference plots were generated using the python package matplotlib[101]. Curve fitting and statistical tests were performed in python with SciPy[102].

### Quantitative editing assays with diploid reporter yeast

Editing of our diploid SSA reporter strain was initiated by co-delivering native Cas9 or Cas9-scMCP backbones with Wt or MT3 PID inserts as described above for transformation-associated editing of haploid SSA reporter strains with a single sgRNA. However, transformants were not plated immediately after the 10 min incubation in CaCl$_2$; rather, half the resuspended mixture (200 μl) was inoculated into 20 mL SC−Leu−Ura broth to provide selection for cells which maintain both Cas9 (assembled in vivo) and sgRNA episomes, as well as time for steady-state segregation of the newly assembled Cas9 episomes. These 20 mL cultures were grown for 24 h at 30 °C with rotation on a roller drum and then subcultured 1:20 in a fresh 20 mL volume of the same media, either lacking β-E or supplemented with β-E at 15 nM or 1000 nM. Subcultures were grown at 30 °C with rotation on a roller drum until $D_{600\ nm}$ ~ 1.0 (12–17 h), pelleted by centrifugation at 3000 × *g* for 3 min, washed once with resuspension in an equal volume of sterile ddH$_2$O, and then re-pelleted and frozen at −20 °C for downstream gDNA isolation by bead beating and phenol-chloroform extraction (25:24:1 phenol:chloroform:isoamyl alcohol). Bead beating was carried out at 4 °C on a FastPrep-24 5 G machine from MP Biomedicals using their 2 mL tubes pre-filled with Lysing Matrix Y; eight cycles of 15-second bead beating was performed with 30 s pauses between each cycle. For the phenol-chloroform extraction, treatment with RNAse A (Thermo Scientific) at a final concentration of 30 mg/mL was performed for 15 min at 37 °C before the final 70% ethanol precipitation step. In parallel, aliquots were withdrawn from the post−24 h transformant cultures to determine their *LYS2*[+] fractions immediately prior to the 12–17 h subculture step. These aliquots were first pelleted by centrifugation at 2103 × *g* for 4 min, washed twice in an equal volume of sterile ddH$_2$O with resuspension and re-pelleting with the same centrifugation conditions, and then resuspended in a final volume of 1 mL ddH$_2$O before serially diluting and spotting on plates. Plates were incubated 36 h at 30 °C before immediately photographing and quantifying CFUs as described for haploid editing assays, except only 12 h were allowed at benchtop before recounting.

To analyze allele frequencies at *LYS2* loci within the above gDNA preparations, targeted deep sequencing using an Illumina MiSeq (v2 reagent kit) was performed on amplicons that spanned a 230 bp region with the protospacer and allele-specific point mutation. These were amplified from unedited alleles in a single round of PCR (25 cycles) with GoTaq Green and unique combinations of primers oGG1246-oGG1254 (Supplementary Data 4) for multiplexed sequencing with dual indexing in a single run. Demultiplexed paired-end (230 × 2) reads were analyzed with IDT's rhAmpSeq CRISPR Analysis Tool (CRISPAltRations 1.1.0) using a modified protocol to quantify allele frequencies rather than Cas9 editing per se. Briefly, the 230 bp target region at Allele 1 was provided to the tool as 'Amplicon' sequence, while the same region but with the Allele 2 point mutation was provided as 'Donor' sequence. In addition, the 'Guide RNA' sequence provided to the tool was a mock spacer that overlapped the point mutation position, which was thus the fourth bp upstream of the PAM. Using these parameters, the 'Percent Unedited' output corresponds to the percentage of reads containing the Allele 1 sequence, while the 'Percent Perfect HDR' output corresponds to the percentage of reads containing the Allele 2 sequence. These output values summed to ≥ 98.5% for each sample analyzed, confirming that the majority of data were considered in our allelic ratio calculations. Approximately 0.204–0.473 million merged reads were analyzed per sample.

### Plating assay to measure colony sizes

To obtain liquid cultures for plating, 18 × 150 glass tubes containing 5 mL SC−Leu or SC−Leu−Ura broth were inoculated with single colonies and grown for 22 h on a roller drum at 30 °C. SC−Leu−Ura was used for strains with the sgRNA plasmid while SC−Leu was used for strains without. Once grown, cultures were diluted in sterile ddH$_2$O and then

plated on solid media of the same type to obtain ~150–400 colonies per plate. Plates were photographed directly from above after 47 h growth at 30 °C and colonies were then analyzed as 2-dimensional objects using custom python scripts to quantify colony areas. This was achieved using the python package scikit-image: colonies were segmented by the Otsu thresholding algorithm and touching colonies were separated by the watershed method. Objects detected near plate edges were excluded to avoid measuring artifacts. Swarm plots were generated using the python package seaborn and $t$ tests were performed with SciPy[99].

## MNase-seq procedures and analyzes

Experiments employed a *pZEVi-LYS2*[+] haploid strain (yGWG538 in Supplementary Data 2) harboring an intact *LYS2*[+] ORF, the native Cas9[Wt] plasmid (pGG541), and the single *EGFP* sgRNA plasmid (pGG197) which is non-targeting in this background. These features were chosen (1) to ensure standard mRNA quantifications performed in parallel (see qRT-PCR section below) by avoiding nonsense-mediated decay expected in targeting strains and (2) to emulate the plasmid conditions of our other experiments without interfering with growth. The following procedures were adapted from MNase-seq methods described previously[103,104]. Liquid starter cultures grown for 24 h at 30 °C were obtained by inoculating 5 mL SC–Leu–Ura media from single colonies and then subcultured (1:70) into 500 mL flasks with 100 mL of the same media, either lacking β-E or supplemented at 15 nM and 1000 nM. These were grown at 30 °C with shaking (220 RPM) until $D_{600\,nm}$ ~ 1.0 ( ~ 7 h) and then crosslinked by adding formalin (1% final concentration) and incubating at 25 °C with gentle shaking (110 RPM) for 15 min. After adding 5 mL glycine (125 mM final concentration) to quench formalin for 5 min at 25 °C with gentle shaking (110 RPM), cultures were pelleted by centrifugation at 4 °C for 5 min at 3000 × $g$, washed twice with resuspension in equal volumes of ice-cold sterile ddH$_2$O, and then pelleted again in 1.5 mL tubes for snap-freezing in liquid nitrogen and storing at −80 °C. Pellets were thawed on ice and resuspended in spheroplasting buffer (1.2 M sorbitol, 100 mM potassium phosphate, 1 mM CaCl$_2$, 0.5 mM β-mercaptoethanol, pH 7.5) with 1 mg/mL of 100 T Zymolyase. Cells were spheroplasted at 37 °C for 30 min and spheroplasts were collected by centrifugation and washed twice with 1 mL of spheroplasting buffer. Spheroplasts were divided into four 1.5 mL tubes, pelleted by centrifugation, and resuspended in MNase digestion buffer (1 M Sorbitol, 50 mM NaCl, 10 mM Tris-HCL (pH 7.4), 5 mM MgCl$_2$, 1 mM CaCl$_2$, 0.5 mM spermidine, 0.075% NP-40, add 1 mM β-mercaptoethanol) with 20 U of MNase (Thermo Scientific Cat. EN0181). Chromatin was digested with MNase at 37 °C for 15, 30, 45, and 60 min and reactions were stopped by addition of EDTA to a final concentration of 30 mM. Cross-links were reversed by addition of SDS (to 0.5% final) and proteinase K (0.5 mg/ml final) and incubated at 37 °C for 1 h followed by 65 °C for 2 h. Digested DNA was extracted using 25:24:1 phenol:chloroform:isoamyl alcohol and precipitated with 70% isopropanol. To remove RNA, the mixture was then resuspended in TE buffer with RNASe A (6 Kunitz units) and incubated at 37 °C for 1 h. Finally, DNA was purified with the Zymo Clean and Concentrator kit. MNase digestions were assessed by running on an agarose gel with ethidium bromide staining to confirm mono-nucleosome fragments between 100-200 bp. A multiplexed Illumina sequencing library was then prepared from samples digested for 15 and 60 min through direct adapter ligation using the automated KAPA general DNA-seq library preparation kit. Paired-end (51 × 2) sequencing of this library on an Illumina NovaSeq 6000 with a SP100 reagent kit yielded approximately 40 million reads per sample after demultiplexing. Demultiplexed reads were first analyzed with Trimmomatic (v0.39) processing to remove sequencing adapters and then with FastQC (v0.11.4) to assess read quality. Processed reads were then aligned to a custom reference of chromosome II (chr.II) containing *pZEVi-LYS2*[+], using the Burrows Wheeler aligner (BWA) mem algorithm (v0.7.7). Aligned reads

were then used as input for nucleosome analysis using the DANPOS (v2) pipeline[105]. Analyzes of samples digested for 15 and 60 min revealed similar trends but suggested incomplete digestion at 15 min; hence, only 60 min data are presented. We observed a total of 4,670 nucleosomes across chr.II with ~ 1% of these nucleosomes showing statistically significant changes in occupancy levels when comparing between 0 nM β-E and 15 nM or 1000 nM, of which about half were from the *LYS2*[+] locus (Supplementary Data 1; $p$ value < 0.01; FDR-corrected Poisson tests performed by DANPOS). Tukey's tests for significant Log2FC differences across the *LYS2*[+] ORF (Supplementary Fig. 7f) were performed using GraphPad Prism software.

## qRT-PCR and Fold Change calculations for *pZEVi-LYS2* induction

To purify total RNA for reverse transcription, 1.3 mL aliquots were withdrawn from MNase-seq cultures immediately prior to formalin addition, pelleted at 4 °C, washed once with resuspension in 1.3 mL ice-cold autoclaved ddH$_2$O, then re-pelleted at 4 °C for −80 °C storage until further processing. RNA was purified using the RNeasy Mini kit (Qiagen) and an in-solution DNase I (Qiagen) treatment prior to final cleanup. cDNA was synthesized using SuperScript IV Reverse Transcriptase (Invitrogen) according to the manufacturer's recommendations for a 20 μl reaction with 1 μg total RNA and random hexamers (IDT). qPCRs were performed with gene-specific primers (each at 500 nM) from IDT and 2.5 μL of the diluted cDNA (1:5 dilution) in a 10 μL reaction with Fast SYBR Green Master Mix (Life Technologies) and a LightCycler® 480 Real-time PCR System (Roche) according to the manufacturers' protocols. Primers used for the *TAF10* housekeeping gene or *LYS2* target gene are listed in Supplementary Data 4. Raw Ct values were calculated using the second derivative method. ΔCt values were calculated relative to the *TAF10* levels in each sample; ΔΔCt values were calculated relative to the *LYS2* levels in replicate 3 of the 0 nM samples. Ultimately, the plotted Fold Changes in *LYS2* RNA levels were calculated as Fold Change = $2^{-\Delta\Delta Ct}$.

## ChIP-qPCR and ChIP-seq procedures

Experiments employed *pZEVi* yeast strains derived from yGWG385 (Supplementary Data 2) via co-transformation with various (d)Cas9/sgRNA plasmid combinations (Supplementary Data 1). yGWG385 contains the *pZEVi-LYS2::boxB*[1x](Upstr.) non-SSA target insertion used in donor-dependent editing assays; to prevent cleavage-dependent growth perturbations when inducing *pZEVi* in this strain background (Supplementary Figs. 6–7), we used a non-targeting sgRNA with Cas9[NG] and Cas9[NG]-2xN, or alternatively, a targeting sgRNA with non-cleaving dCas9[NG] and dCas9[NG]-2xN variants. ChIP procedures were adapted from existing protocols[106–109] described for *S. cerevisiae* that typically recommend isolating chromatin from crosslinked cells via bead beating.

To obtain crosslinked cells for ChIP, we followed all MNase-seq procedures described above for culturing from single colonies, subculturing with or without β-E induction, formalin cross-linking, glycine quenching, pellet washing, and snap-freezing for storage at −80, except that ice-cold sterile PBS (1X) was used for the two washes in place of ddH$_2$O. Unless otherwise noted, all subsequent steps for ChIP were performed on ice or at 4 °C using pre-chilled tubes, solutions, and equipment. After thawing on ice, pellets were resuspended in 1.4 mL Lysis Buffer (50 mM HEPES-KOH pH 7.5, 150 mM NaCl, 1 mM EDTA pH 8.0, 1% (v/v) Triton X-100, 0.1% (w/v) sodium deoxycholate, 0.1% (w/v) SDS) and transferred to 2 mL screwcap tubes pre-filled with 0.5 mm yttria-stabilized zirconium oxide beads (Lysing Matrix Y from MP Biomedicals) for bead beating. Our Lysis Buffer is identical to the 'FA lysis buffer' described previously[107] but with the following protease inhibitors freshly added before use: yeast-optimized Protease Inhibitor Cocktail from Sigma (P8215) diluted 1:200, and dissolvable cOmplete Mini EDTA-free Protease Inhibitor Cocktail tablets from Roche. Bead beating was carried out at 4 °C on a FastPrep-24 5 G machine from MP

Biomedicals with seven 30-second cycles set to 8 m/sec (QuickPrep adapter) and 3 min pauses on ice between each cycle. To separate chromatin lysates from the beads, a tube-in-tube setup was used to centrifuge lysates from screwcap tubes into 2 mL LoBind (Eppendorf) tubes below, after cutting off the LoBind tube caps. First, the capless 2 mL LoBind tube was inserted into the bottom of a 14 mL round-bottom falcon tube. Second, the screwcap tube was inverted and punctured once at the bottom with a disposable 25 G 5/8" needle. Third, the screwcap tube was placed upright on the LoBind tube within the falcon and centrifuged once at 200 × g for 2 min and again at 400 × g for 2 min; in some cases, a second 2 min spin at 400 × g was needed to finish collecting the lysate below. Lysates were then pipette mixed before transferring to fresh 2 mL LoBind tubes. A total of 630 μl lysate from each sample was next aliquoted for sonication in three 1.5 mL TPX plastic tubes from Diagenode (210 μl lysate per tube). Sonication was performed with a Diagenode Bioruptor Plus device set to 'High' for 21 consecutive cycles of 30 s ON and 30 s OFF and a Bioruptor Water cooler/pump unit set to 2 °C. Lysates were re-pooled from sonication tubes into single 1.5 mL LoBind (Eppendorf) tubes and centrifuged at 16,000 × g for 15 min to pellet residual cellular debris, and supernatants containing sonicated chromatin were then transferred to fresh 1.5 mL LoBind tubes for downstream immunoprecipitation (IP) with magnetic Dynabeads pre-conjugated to antibody. ChIP-grade anti-HA antibody from Abcam (ab9110) was covalently conjugated to M-270 Epoxy Dynabeads beforehand at a ratio of 5 μg per 100 mg beads, stored at 4 °C, and finally blocked at r.t. immediately before use, in each case by following manufacturer protocols. After blocking, Dynabeads were resuspended in Lysis Buffer (120 μl per 100 mg beads) and volumes containing 100 mg beads were mixed into each sonicated sample to carry out IPs with rotation at 4 °C for 20 h. Prior to mixing with Dynabeads, a 20ul aliquot of the 'input' chromatin was withdrawn from each sonicated sample and stored at 4 °C until the de-crosslinking step carried out alongside pull-down samples the next day.

To isolate DNA from input and pull-down samples, Dynabead-bound pull-downs were first washed at r.t. several times using a magnetic stand and the following buffers: twice with Lysis Buffer, once with High-salt Buffer, once with LiCl/DOC Buffer, and once with TE Buffer (10 mM Tris pH 8.0, 1 mM EDTA pH 8.0). Our High-salt Buffer is identical to Lysis Buffer (see above) except with 500 mM NaCl in place of 150 mM NaCl, and only the liquid Protease Inhibitor Cocktail from Sigma (P8215) was added before use. Our LiCl/DOC Buffer (10 mM Tris pH 8.0, 0.25 M LiCl, 1 mM EDTA pH 8.0, 0.5% (v/v) Nonidet P-40, 0.5% (w/v) sodium deoxycholate) is identical to the 'ChIP wash buffer' described previously[107] except it was supplemented with Protease Inhibitor Cocktail from Sigma (P8215) before use. Aliquots of Lysis/High-salt Buffers were equilibrated to r.t. before use. After resuspension in each buffer (700 μl per buffer), Dynabeads were incubated with rotation for 3-5 min at r.t. and then pulse-centrifuged before handling on the magnet. To elute pull-downs, Dynabeads were resuspended in 100 μl TES (50 mM Tris pH 8.0, 10 mM EDTA pH 8.0, 1% SDS) and incubated in a thermomixer for 1 h at 65 °C with light agitation (450 RPM); 95 μl of each eluate was then transferred to fresh LoBind tubes for 18 h de-crosslinking in a thermomixer at 65 °C with agitation (800 RPM). At the same step, input samples were equilibrated to r.t. and mixed with 75 μl TES for 18 h de-crosslinking alongside pull-down samples. De-crosslinked samples were subsequently mixed with 200 μl Buffer TE (to dilute SDS) and 5 μl RNase A/T1 cocktail (2 mg/mL stock) for a 1 h incubation at 37 °C to remove RNA, followed by 10 μl Proteinase K (20 mg/mL stock) for 2 h at 56 °C to remove protein. DNA was then purified using Qiagen MinElute columns. We essentially followed Qiagen's standard protocol for PCR purification but mixed 10 μl sodium acetate into each sample before column loading, performed an extra PE wash, and re-loaded the first eluate (~20 μl) to perform a second elution. After purification, all input samples were analyzed with

an Agilent TapeStation 4150 (D5000 kit) to identify the peak of each fragment length distribution achieved by sonication; peaks ranged from 191-561 bp across samples, with most samples peaking between 200-300 bp.

To quantify pull-down efficiencies by qPCR we used KAPA SYBR FAST (2X Master Mix, Universal kit) and a LightCycler® 480 Real-time PCR System (Roche) according to the manufacturers' protocols, with gene-specific primers (each at 200 nM) from IDT and 1.0 μL of diluted input or pull-down DNA (1:8 dilution to conserve samples) as template in a 10 μL reaction. Primers used for the *LYS2* and *TAF10* amplicons are listed in Supplementary Data 4. Every biological replicate sample was assayed with three technical replicate reactions for each primer set. Raw Ct values were calculated for each technical replicate using the second derivative method, and the mean values from technical triplicates were used to calculate ΔCt values for each primer set by subtracting the mean Ct value of an input sample from the mean Ct value of its corresponding pull-down sample (pull-down − input). Ultimately, the % of input captured by IP was calculated for *LYS10* and *TAF10* as % = 100 × ($2^{-\Delta Ct}$).

For ChIP-seq, a subset of the input and pull-down samples tested by qPCR were also used to prepare libraries for paired-end (51 × 2) sequencing on a single lane of the NovaSeq X Plus (Illumina). All sequencing libraries were prepared using the ThruPLEX DNA-seq kit that can be carried out with as little as 50 pg starting material. Library preparation and sequencing were performed in accordance with manufacturer protocols.

## ChIP-seq analyzes

Raw demultiplexed paired-end reads from each sample were first evaluated for quality using FastQC and then trimmed of low-quality base pairs and adapter sequences from both 5' and 3' ends using the tools Trim Galore (Babraham Bioinformatics) and Trimmomatic. Next, the reads were aligned to their corresponding custom reference sequence using bowtie2 and the results were filtered with samtools to retain only properly paired reads with a minimum mapping quality of 30. The function bamCompare from the package deepTools was used to normalize for read depth (Counts Per Million) and produce a final output as the ratio of IP coverage over input coverage, in both bigwig and bedgraph format. Signals were then plotted across each genomic region of interest using IGV and final plots used for the figures were created with python (matplotlib). Our custom references contained the entire *S. cerevisiae* genome with the corresponding modified chromosomal sequences and episomal Cas9/sgRNA sequences in each strain. All plots represent coverage data that were quantified with a 10 bp binning scheme. Alternatively, to calculate mean coverage ratios across the 20 bp protospacer targeted by dCas9$^{NG}$ and dCas9$^{NG}$-2xN (Supplementary Fig. 11, legend), coverage data were quantified at single-bp resolution.

## Generation of the clonal U2OS.EGFP::5' UTR-*boxB*$^{2x}$ and U2OS.EGFP::Intron-*boxB*$^{2x}$(V1.0) cell lines

All cell lines generated in this study are derivatives of the U2OS.EGFP cell line[18] obtained from Keith Joung's Lab. U2OS.EGFP was derived from U2OS through low-MOI lentiviral transduction to introduce an expression cassette that includes *EGFP-PEST*; the expected sequence of its lentiviral insertion was provided by Toni Cathomen's lab and published previously[19]. Using CRISPR-assisted knock-in, the U2OS.EGFP cell line was modified to contain two *boxB* HTs embedded within the 5' UTR or an intron of its *EGFP-PEST* marker. Briefly, to generate the 5' UTR insertion in U2OS.EGFP::5' UTR-*boxB*$^{2x}$, 2 μg of Cas9 plasmid (pAG5), 666 ng of sgRNA plasmid (pGG771) and 20 pmol of a linear ssDNA repair template (oGG1238) with two phosphothiorate linkages on each end were mixed with 82 μL Nucleofector Solution V and 18 μL of Supplement Solution 1 according to manufacturer's recommendations for the Cell Line Nucleofector Kit V (Lonza). 5 × 10$^5$ U2OS.EGFP parent cells were resuspended in the above mixture and nucleofected

with a Lonza Nucleofector 2b Device (Kit V, Program X-001). The cells were immediately transferred to a 6-well plate containing 2 mL of medium and incubated at standard conditions (37 °C and 5% CO$_2$). Five days post-nucleofection, single cells were sorted based on forward scatter area and width parameters into separate wells of 96-well plates using a BD FACSAria IIu flow cytometer. Wells that showed outgrowth within 7 days post-sorting were genotyped with primers P40 and P41 (Supplementary Data 4) for PCR-based screening of clones carrying the desired knock-in. Out of 26 clones screened, 7 contained an amplicon detectably longer than the parental genotype. These 7 clones were further verified by Sanger sequencing and 6 of 7 were found to contain a perfect insertion, implying an overall efficiency of ~ 23% perfect insertions. A single clone among those 6 was chosen for downstream assays. To generate the intron insertion in U2OS.EGFP::Intron-$boxB^{2x}$(V1.0), transfections were performed using a modified Effectene reagent (Qiagen) protocol as previously described[110] except that the DMEM used for cell culture was also supplemented at ~ 1% each of sodium pyruvate (100 mM stock), GLUTAMAX (100X), and non-essential amino acids (100X) from Gibco. Per transfection, 0.25 μg of Cas9 plasmid (pAG5), 0.1 μg of sgRNA plasmid (pGG443) and 5.5 pmol of a linear ssDNA repair template (oGG1239) was used in 100 μL of buffer EC with a DNA:Enhancer ratio of 1:8 (3.2 μL) and DNA:Effectene ratio of 1:15 (6 μL). The resulting transfection complexes were added to each well in a 6-well plate that was freshly seeded with $2 \times 10^5$ U2OS.EGFP parent cells in 2 mL of media. Seven days post-transfection, single cells were sorted based on forward scatter area and width parameters into separate wells of 96-well plates using a BD FACSAria IIu flow cytometer. Wells that grew out within 7 days post-sorting were genotyped with primers P31 and P32 (Supplementary Data 4) for PCR-based screening of clones carrying the desired knock-in. Out of 91 clones screened, 18 contained an amplicon detectably longer than the parental genotype. Sanger sequencing of the 18 amplicons revealed only one sequence that matched the repair template sequence, except for two unexpected point mutations within the intron's polypyrimidine tract and splice acceptor elements, and the corresponding cell line was isolated for downstream assays conducted in the same medium.

### Human cell culture assays to measure Cas9-mediated indels
To generate indels in the U2OS.EGFP cell line or its $boxB^{2x}$-containing derivatives, Cas9$^{Wt-alt}$ (pAG5) or Cas9-2xN$^{Wt-alt}$ (pGG768) plasmids were co-delivered with sgRNA plasmids (Supplementary Data 3) via nucleofection as described previously for EGFP knockdown assays[19] except that ptdTomato was excluded, media lacked Geneticin (G418 Sulfate), cells were harvested 48 h post-nucleofection for genomic DNA (gDNA) purification rather than flow cytometry, and all three biological replicates were performed on the same day. gDNA was purified from cells in 6-well plates using the DNeasy blood and tissue kit (Qiagen) according to the manufacturer's protocol. For indel detection at the targets of sgRNAs 1 through 3, deep sequencing with an Illumina NextSeq 2000 (300-cycle Low Output P1 reagent kit) was performed on PCR amplicons spanning 297 bp of the protospacer region, and demultiplexed single-read (297 × 1) sequences were analyzed with IDT's rhAmpSeq CRISPR Analysis Tool (CRISPAltRations 1.1.0) using the default parameters for amplicon analyzes. To employ this tool for our single-read data, it was necessary to first generate mock R2 read pair '.fastq' files from the R1 files—with identical but reverse-complemented sequences; then, generate sub-sampled files for downstream analyzes—each with 0.100 million random pairs of reads. Both file processing steps were performed with the Seqtk script for python. Ultimately, > 0.098 million merged reads were analyzed per sample. For indel detection at the sgRNA 4 target, deep sequencing with an Illumina MiSeq (600-cycle v3 reagent kit) was performed on PCR amplicons spanning 297 bp of the protospacer region, and demultiplexed paired-end (297 × 2) reads were analyzed with IDT's rhAmpSeq CRISPR Analysis Tool (CRISPAltRations 1.1.0) using the

default parameters for amplicon analyzes. Approximately 0.690-1.159 million merged reads were analyzed per sample without Seqtk pre-processing. Sequences of the amplified regions provided to the tool as "Amplicons" are included in Supplementary Data 6 along with raw results from the analyzes. PCR amplicons were generated through a single round of GoTaq Green PCR (30 cycles) using unique combinations of primers listed in Supplementary Data 4 for dual-indexed sequencing runs. To facilitate this PCR reaction, the first 10 cycles annealed at 55 °C and the last 20 cycles annealed at 65 °C. Amplified DNA was size-selected (~ 300-500 bp) by gel purification prior to sequencing, accounting for the total amplicon length expected with Illumina adapters (433 bp for unedited genotypes).

### Analysis of cellular fluorescence by flow cytometry
To analyze the mCherry−2xN yeast strains for fluorescence, 1 mL starter cultures were first obtained for each biological replicate as described above in plate reader growth assays, except SC−Ura broth was used. Once grown, 33.3 μl aliquots were withdrawn from starter cultures and analyzed on a Sony SA3800 spectral analyzer after dilution into 500 μl aliquots of PBS + 0.5% FBS on ice. Mean mCherry values were measured in arbitrary fluorescence units from 30,000 events per culture according to the manufacturer's suggested settings for mCherry fluorescence, after gating on a majority population in the expected size range. The gating strategy used for analysis in the SA3800 software is shown in Supplementary Fig. 15. To compare EGFP fluorescence levels in the human U2OS.EGFP line and its U2OS.EGFP::5' UTR-$boxB^{2x}$ derivative, cells were cultured in 6-well plates with Geneticin-free media as described for indel experiments post-nucleofection. Three wells for each line (biological replicates) were seeded in parallel on the same day with $5 \times 10^5$ cells per well. Following 48 h of growth, cells were harvested and analyzed on a Sony SH800 sorter after resuspension in 500 μl aliquots of PBS + 0.5% FBS on ice. Mean EGFP values were measured in arbitrary fluorescence units according to the manufacturer's suggested settings, either for 'all events' (20,000 per sample) or for 'singlet events' (at least 19,000 per sample) enriched by gating. The gating strategy used for analysis in the SH800 software is shown in Supplementary Fig. 16.

### Statistics & reproducibility
GraphPad Prism, matplotlib (python), DANPOS v2 (python), IGV, or Microsoft Excel were used to generate all quantitative plots in this work before manually compiling and annotating them in Adobe Illustrator. Except where noted otherwise, $p$ values were calculated with GraphPad Prism or python (SciPy) software using unpaired, two-tailed $t$ tests with Welch's correction, Gaussian distributions assumed, and no correction for multiple comparisons. All assays were performed with at least three biological replicates except where noted otherwise, and biological replicates were performed on separate days unless noted otherwise. No statistical method was used to predetermine experimental sample sizes. No samples were excluded from data analyzes.

### Reporting summary
Further information on research design is available in the Nature Portfolio Reporting Summary linked to this article.

## Data availability
Relevant data supporting the findings of this study are available in the published article, its supplementary files, and the Sequence Read Archive of the NCBI (BioProject accession codes: PRJNA1149169 and PRJNA1149094). Source data are provided with this paper.

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

## Acknowledgements

We thank former Noyes lab members David Ichikawa and Jeff Spencer, as well as Ran Brosh and other current or former members of the Boeke and Noyes labs, for sharing helpful discussions and resources. Paolo Mita provided extensive guidance on ChIP procedures and the pPM193 plasmid template from the Boeke lab for amplifying the Wt TetR PCR fragment of scTetR. Raquel Ordonez (Maurano lab) and Mariya London (Mucida lab alumnus) provided additional advice on ChIP procedures. Mike Shen (Boeke lab alumnus) designed the pMS101 plasmid in collaboration with the Noyes lab. Makiha Fukuda of the Boeke lab provided *TAF10* primers for q(RT-)PCR. We also thank Addgene for providing plasmid templates to amplify a recoded TetR PCR fragment and an isogenic RevTetR PCR fragment: the pCW57.1-MAT2A plasmid[111] (#100521; RRID:Addgene_100521) and the pCAG-TetON-3G plasmid[112] (#96963; RRID:Addgene_96963), respectively, gifts from David Sabatini's lab and Elena Cattaneo's lab. The BPK1520 plasmid[18] acquired through Addgene (#65777; RRID:Addgene_65777), as well as the U2OS.EGFP cell line[18], were gifts from Keith Joung's lab. We gratefully acknowledge NYU Medical Center's Research Support Service team for preparation of various reagents and yeast media used in this work, the Genome Technology Center (RRID: SCR_017929) for low-complexity amplicon sequencing on MiSeq/NextSeq 2000 platforms, as well as library preparation and sequencing for MNase-seq (NovaSeq 6000) and ChIP-seq (NovaSeq X Plus), and the IDT rhAmpSeq CRISPR Analysis Tool team for their services and tech support. We are also grateful for thoughtful discussions with Timothee Lionnet and other members of his lab, as well as Rachel Leicher from Shixin Liu's lab. G.W.G. was supported by an NRSA Postdoctoral Fellowship (F32GM137482) and an NIH K99 Pathway to Independence Award (K99GM147604), both from the NIGMS, as well as a Centers of Excellence in Genomic Science award (RM1HG009491) to J.D.B. from the NHGRI. Additional funding support was provided from NIGMS awards R01GM118851 and R01GM133936 to M.B.N.

## Author contributions

G.W.G. conceived the study. G.W.G. designed experiments with help from J.D.B., M.A.B.H., L.K., and M.K. G.W.G., D.K. B., M.A.B.H., L.K., M.K., and K.O. generated biological reagents and/or conducted experiments. S.K. analyzed high-resolution growth curve data, colony size data, and ChIP-seq data. M.A.B.H. analyzed MNase-seq data. M.A.A. ran Seqtk processing for single-read NextSeq data files. S.A., J.D.B., D.F., M.A.A., and M.B.N. contributed other resources and/or technical advice. J.D.B. and M.B.N. provided funding. G.W.G. wrote the paper with help from J.D.B., M.A.B.H., L.K., S.K., and M.K.

## Competing interests

Jef Boeke is a Founder and Director of CDI Labs, Inc., a Founder of and consultant to Opentrons LabWorks/Neochromosome, Inc, and serves or served on the Scientific Advisory Board of the following: CZ Biohub New York, LLC, Logomix, Inc., Modern Meadow, Inc., Rome Therapeutics, Inc., SeaHub, Seattle, WA, Tessera Therapeutics, Inc. and the Wyss Institute. Marcus Noyes is a founder of TBG Therapeutics, Inc. New York University filed a provisional patent application (No. 63/582,731) for findings described in this work, with Gregory Goldberg, Marcus Noyes, and Jef Boeke listed as inventors. The other authors declare no competing interests.
