## [Transparent Peer Review file · Nature Communications]

Engineered transcription-associated Cas9 targeting in eukaryotic cells

Corresponding Author: Dr Gregory Goldberg

This manuscript has been previously reviewed at another journal. This document only contains information relating to versions considered at Nature Communications. Mentions of the other journal have been redacted.

Version 0:

Reviewer comments:

Reviewer #1

(Remarks to the Author)

The authors have sufficiently addressed the reviewers' comments, and I agree the move to Nature Communications was a good fit given the novelty of the technique and the extent to which the technique has been demonstrated and can be applied.

Reviewer #2

(Remarks to the Author)

Goldberg et al. present a revised version of their TraCT manuscript, now being assessed in more appropriate and fitting [redacted]. The revisions and rebuttal are noted, with noteworthy expansion of the cited literature, contextualization of the findings in light of relevant published studies, and clarifications of the novelty of the study in general and regarding relevant efficiency in particular. It is also important to note the expanded narrative on needed improvements, limitations and technical constraints, for the benefit of the readership. I note the rebuttal is generally agreeable, and my concerns have been addressed, notably regarding mammalian cells and eukaryotic context, general applicability and significance of the results, and conclusiveness of the statements presented.

I do think the editorial team should ensure all statistically significant differences are labelled in relevant figures (e.g. Fig1 bcef and Fig2 seem to be missing entries).

Reviewer #3

(Remarks to the Author)

The authors have added substantial new data to support several claims, specifically regarding whether the method provides an accurate readout of transcriptional activity and the impact of the inserted RNA hairpins on native transcriptional processes. They have also addressed concerns about the technical challenges of hairpin insertion and applications of this method through textual revisions. The inclusion of the ChIP experiment provides direct evidence supporting their major claim, which strengthens the manuscript. This review finds the revisions satisfactory. The revised manuscript is suitable for publication.

Point-by-point response to Reviewers' comments
(Authors' responses in blue text)

Reviewer Comments:

Reviewer #1 (Remarks to the Author):

The authors have sufficiently addressed the reviewers' comments, and I agree the move to Nature Communications was a good fit given the novelty of the technique and the extent to which the technique has been demonstrated and can be applied.

We thank this reviewer for their highly constructive and insightful critiques throughout the review process, and for explicitly affirming that we have addressed each of the reviewers' comments.

Reviewer #2 (Remarks to the Author):

Goldberg et al. present a revised version of their TraCT manuscript, now being assessed in more appropriate and fitting [redacted]. The revisions and rebuttal are noted, with noteworthy expansion of the cited literature, contextualization of the findings in light of relevant published studies, and clarifications of the novelty of the study in general and regarding relevant efficiency in particular. It is also important to note the expanded narrative on needed improvements, limitations and technical constraints, for the benefit of the readership. I note the rebuttal is generally agreeable, and my concerns have been addressed, notably regarding mammalian cells and eukaryotic context, general applicability and significance of the results, and conclusiveness of the statements presented.

We thank this reviewer for critically assessing our study, its novelty, the manner in which we presented the findings, and the appropriateness of the revised manuscript for the current journal. Their comments, along with comments from the other reviewers, prompted key improvements to the manuscript.

I do think the editorial team should ensure all statistically significant differences are labelled in relevant figures (e.g. Fig1 bcef and Fig2 seem to be missing entries).

It should be clarified that statistical testing was not performed for every result described in this work; accordingly, **no claims of statistical significance were made when describing data (such as those mentioned by the Reviewer) that were not tested for statistical significance.**

Nevertheless, **Source Data are now provided for those figures**, so it will be entirely possible for the interested reader to perform their own statistical testing on those data as needed.

Reviewer #3 (Remarks to the Author):

The authors have added substantial new data to support several claims, specifically regarding

whether the method provides an accurate readout of transcriptional activity and the impact of the inserted RNA hairpins on native transcriptional processes. They have also addressed concerns about the technical challenges of hairpin insertion and applications of this method through textual revisions. The inclusion of the ChIP experiment provides direct evidence supporting their major claim, which strengthens the manuscript. This review finds the revisions satisfactory. The revised manuscript is suitable for publication.

We thank this reviewer for their consistently critical reading of the manuscript and the numerous insightful comments they provided at each step of the review process.